# Masked Diffusion Models as Energy Minimization

**Sitong Chen**[1,2,3], **Shen Nie**[1,2,3], **Jiacheng Sun**[4],
**Zijin Feng**[4], **Zhenguo Li**[4], **Ji-Rong Wen**[1,2,3], **Chongxuan Li**[1,2,3*]
[1] Gaoling School of Artificial Intelligence, Renmin University of China
[2] Beijing Key Laboratory of Research on Large Models and Intelligent Governance
[3] Engineering Research Center of Next-Generation Intelligent Search and Recommendation, MOE
[4] Huawei Noah's Ark Lab
{chensitong0809,nieshen,jrwen,chongxuanli}@ruc.edu.cn;
{sunjiacheng1,zijin.feng,Li.Zhenguo}@huawei.com

## Abstract

We present a systematic theoretical framework that interprets masked diffusion models (MDMs) as solutions to energy minimization problems in discrete optimal transport. Specifically, we prove that three distinct energy formulationskinetic, conditional kinetic, and geodesic energyare mathematically equivalent under the structure of MDMs, and that MDMs minimize all three when the mask schedule satisfies a closed-form optimality condition. This unification not only clarifies the theoretical foundations of MDMs, but also motivates practical improvements in sampling. By parameterizing interpolation schedules via Beta distributions, we reduce the schedule design space to a tractable 2D search, enabling efficient post-training tuning without model modification. Experiments on synthetic and real-world benchmarks demonstrate that our energy-inspired schedules outperform hand-crafted baselines, particularly in low-step sampling settings.

## 1 Introduction

Masked diffusion models (MDMs) [4, 8, 34, 44, 36, 38] have emerged as a powerful class of generative models for discrete data. By reversing a stochastic masking process, MDMs iteratively generate sequences through a series of unmasking steps, guided by learned denoising functions. This simple yet flexible architecture has shown promising empirical performance across text generation [43], protein generation [39, 40], and image generation [15, 23, 47].

Despite their empirical success, the underlying principles that govern the sampling efficiency of MDMsparticularly in few-step regimesremain poorly understood. Existing works typically adopt manually designed mask schedules (e.g., linear or sine) without theoretical justification. In contrast, in continuous domains [27, 22, 25] and discrete flow matching [37], recent work has drawn deep connections between diffusion models and optimal transport, motivating questions such as: Can MDMs be similarly understood through a similar lens? Can we characterize optimal sampling schedules in a principled way? How do these schedules relate to the geometry of the underlying probability space?

This paper answers these questions by establishing a theoretical framework that interprets MDMs as minimizing energy functionals over discrete probability flows (DPFs). We prove that three natural formulations of transport costkinetic energy, conditional kinetic energy, and geodesic energyare equivalent under the structure of MDMs. More importantly, we show that MDMs minimize these energies when the mask schedule $\alpha_t$ is coupled to a geometric interpolation schedule (also the weight function in the energeis) $\gamma_t$ via a simple closed-form relation: $\alpha_t^\star = \sin^2(\frac{\pi}{2}\gamma_t)$. This result unifies

---

*Correspondence to Chongxuan Li.

39th Conference on Neural Information Processing Systems (NeurIPS 2025).

seemingly disparate formulations and reveals that MDMs not only follow geodesics on the probability simplex, but also implicitly optimize sampling rate matrices despite its structural constraints.

Building on this insight, we propose an efficient parameterization of schedule functions via the cumulative distribution function (CDF) of Beta distributions. This reparameterization reduces the high-dimensional schedule design problem to a 2-dimensional scalar search, enabling task-adaptive tuning with minimal overhead. We validate our theory through extensive experiments on both synthetic and large-scale real-world benchmarks, including language, code, and mathematical reasoning tasks. Our results demonstrate that energy-inspired schedules outperform commonly used manually designed schedules in few-step sampling settings for certain tasks.

In summary, our contributions are:

- We establish a theoretical framework that interprets MDM as optimal transport processes, and prove the equivalence of three distinct energy formulations.
- We derive a closed-form condition for energy-optimal mask schedules, showing that MDMs minimize sampling cost under this condition.
- We introduce a Beta-CDF parameterization that enables efficient and task-adaptive schedule tuning in a 2-dimensional space.
- We empirically validate our theory across synthetic and real-world benchmarks, demonstrating consistent improvements in few-step sampling performance.

## 2 Background

### 2.1 Discrete Probability Flows and Masked Diffusion Models

**Discrete Probability Flows (DPFs).** DPFs [16, 14, 32] define continuous-time transformations over structured distributions in finite state spaces. In text generation tasks, a state $z = (z^1, ..., z^n) \in \mathcal{D}^n$ typically denotes a token sequence of fixed length $n$ drawn from a vocabulary $\mathcal{D}$ of size $d$. Formally, DPFs introduce a family of parameterized distributions $(p_t(z))_{t \in [0,1]}$ that interpolate between a tractable base distribution $p_0(z)$ (e.g., a uniform distribution) and a target distribution $p_1(z)$.

A DPF is governed by a time-dependent transition rate matrix $(Q_t)_{t \in [0,1]}$,[2] which specifies transition probabilities between states. However, the reverse implication does not hold: different rate matrices can induce the same DPF [32]. We will omit the range of the time index $t \in [0, 1]$ for brevity.

**Masked Diffusion Models (MDMs).** MDMs [4, 8] represent a subclass of DPFs built upon absorbing Markov chains. These models capture two temporal processes: a *masking* process (operating backward in time, $t = 1 \rightarrow 0$) and its reverse, the *unmasking* process ($t = 0 \rightarrow 1$) as follows.

$$\text{Masking process:} \quad \overleftarrow{Q_t}(x, z) = \begin{cases} \sigma_t & z \leftarrow x \\ 0 & \text{otherwise} \end{cases} \quad (z \neq x), \tag{1}$$

$$\text{Unmasking process:} \quad Q_t(z, x) = \overleftarrow{Q_t}(x, z)\frac{p_t(x)}{p_t(z)} = \begin{cases} \sigma_t \frac{p_t(x)}{p_t(z)} & z \rightarrow x \\ 0 & \text{otherwise} \end{cases} \quad (x \neq z), \tag{2}$$

where transitions in the masking process involve single-token masking operations: from $x$ to $z$ where a token is replaced by a special mask token [M], i.e., $z = (z^1, ..., z^i = [M], ..., z^n) \leftarrow x = (z^1, ..., x^i \neq [M], ..., z^n)$. The unmasking process reverses this transition.

This absorbing structure yields a closed-form *conditional probability flow*, realized as independent per-token interpolation between the data and mask states:

$$p_{t|1}(x^i|x_1) = (1 - \alpha_t)\delta_{[M]}(x^i) + \alpha_t \delta_{x_1^i}(x^i), \tag{3}$$

where the *mask schedule* $\alpha_t \in [0, 1]$ is a smooth, strictly increasing function satisfying $\alpha_0 = 0$ and $\alpha_1 = 1$. It determines the progression of masking over time. The schedule relates to the rate $\sigma_t$ via:

$$\alpha_t = \exp\left(-\int_t^1 \sigma_s \, ds\right), \tag{4}$$

---

[2]We adopt the MDM terminology rather than the velocity functions used in the flow matching literature.

as shown in existing work [44, 38]. Existing methods model the likelihood ratio $\frac{p_t(x)}{p_t(z)}$ in Eq. (2)also known as the *concrete score* [19, 26] or its equivalent formulations [38, 36, 44]and optimize evidence lower bounds (ELBO) of the log-likelihood [4]. In particular, the ELBO is invariant to the choice of $\alpha_t$ [12, 38, 36] by change of variables. We provide proofs for Eq. (4) in Appendix C.1 and the invariance of ELBO in Appendix C.2 for completeness.

## 2.2 Kinetic and Conditional Kinetic Energy

Kinetic energy [2, 13, 27, 37] provides a principled framework for quantifying the transport cost of probability flows, forming an optimization objective that yield improved sampling trajectories.

**Definition 2.1** (Weighted kinetic energy). *Given a weight function $\gamma_t$ and a DPF $p_t$ governed by rate matrix $Q_t$, the* weighted kinetic energy *is defined as*

$$\mathcal{E}_k(p_t, Q_t; \gamma_t) = \mathbb{E}_{t, p_t(z)} \sum_{x:x \neq z} \frac{1}{\dot{\gamma}_t p_t(x)} Q_t(z, x)^2, \tag{5}$$

*where $\dot{\gamma}_t$ denotes the temporal derivative of $\gamma_t$.*

The function $\gamma_t$ is a smooth, strictly increasing schedule satisfying the boundary conditions $\gamma_0 = 0$ and $\gamma_1 = 1$. Different choices of $\gamma_t$ implement various temporal weighting schemes for the DPF. This generalizes the unweighted kinetic energy [37], where $\gamma_t = t$, and facilitates our theoretical analysis.

The quadratic dependence on the transition rates $Q_t(z, x)$ in Eq. (5) mirrors the classical velocity-squared form of kinetic energy. As $Q_t$ controls sampling behavior, minimizing $\mathcal{E}_k$ seeks minimal-cost sampling paths between distributions [24, 22, 25], a central goal in efficient generative modeling and few-step sampling [2, 13, 27].

However, direct computation of $\mathcal{E}_k$ is generally intractable, owing to the absence of closed-form solutions for $p_t$ in most practical cases. To address this, we introduce a conditional surrogate.

**Definition 2.2** (Weighted conditional kinetic energy). *Let $p_{t|1}$ denote a conditional flow governed by the conditional rate matrix $Q_{t|1}$, which satisfies the marginal consistency condition:*

$$Q_t(z, x) = \sum_{x_1} Q_{t|1}(z, x|x_1) \, p_{1|t}(x_1|z). \tag{6}$$

*The weighted conditional kinetic energy with weight function $\gamma_t$ is defined as*

$$\mathcal{E}_c(p_{t|1}, Q_{t|1}; \gamma_t) = \mathbb{E}_{t, p_1(x_1), p_{t|1}(z|x_1)} \sum_{x:x \neq z} \frac{1}{\dot{\gamma}_t p_{t|1}(x|x_1)} Q_{t|1}(z, x|x_1)^2. \tag{7}$$

Definition 2.2 extends Definition 2.1 to the conditional setting, in line with analogous energy formulations in the continuous domain [27]. While the two energies differ in general, they are equivalent under the MDM framework (see Sec. 3.1).

## 2.3 MDM as Geodesic Curve

To understand the relationship between MDM and kinetic energy from a geometric perspective, we draw on the geodesic interpretation [42]. This view emerges from a key geometric embedding: for each token, the conditional distribution $p_{t|1} = (p_{t|1}^1, ..., p_{t|1}^D) \in \Delta^{D-1}$ can be mapped isometrically onto the unit sphere via square-root parameterization:

$$y_t = \left(\sqrt{p_{t|1}^1}, ..., \sqrt{p_{t|1}^D}\right) \in \mathbb{S}^{D-1} = \{y : \sum_i (y^i)^2 = 1\}. \tag{8}$$

Through this embedding, [42] proved geometrically that the per-token MDM conditional flow Eq. (3) corresponds to geodesic motion on $\mathbb{S}^{D-1}$that is, movement along the great circle connecting the masked initial state $y_0$ and the target distribution $y_1$. The *interpolation schedule* $\gamma_t$ (a smooth increasing function satisfying $\gamma_0 = 0$, $\gamma_1 = 1$) governs the temporal progression, and its derivative $\dot{\gamma}_t$ represents instantaneous velocity. We retain the $\gamma_t$ notation from Definition 2.1, as it also serves as

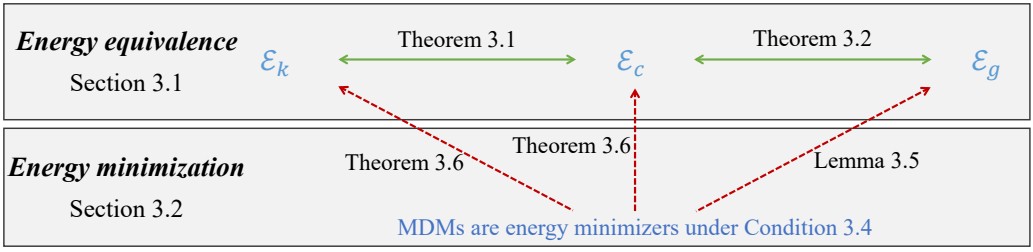

Figure 1: **Illustration of the theoretical results of this paper.**

the weight function for energy minimization as detailed in Sec. 3. A more comprehensive and intuitive introduction to the geodesic curve and its relationship with MDM is provided in Appendix B.

This geometric framework unifies MDM sampling dynamics with energy minimization principles, since geodesics simultaneously minimize both path length and energy functionals [1]. To formalize this link, we define the *weighted geodesic energy*, which MDMs minimize under suitable scheduling.

**Definition 2.3** (Weighted geodesic energy). *Given a weight function $\gamma_t$ and a conditional flow $p_{t|1}$ governed by a conditional rate matrix $Q_{t|1}$, the weighted geodesic energy is defined as:*

$$\mathcal{E}_g(p_{t|1}; \gamma_t) = \sum_{i=1}^{n} \mathbb{E}_{t, p_1(x_1), p_{t|1}(z^i|x_1)} \frac{4}{\dot{\gamma}_t p_{t|1}(z^i|x_1)} \dot{y}_{t|1}(z^i|x_1)^2, \tag{9}$$

*where $y_{t|1}(z^i|x_1) := \sqrt{p_{t|1}(z^i|x_1)}$ is derived from the embedding in Eq. (8).*

The term $\dot{y}^2$ in $\mathcal{E}_g$ again reflects the velocity-squared form of kinetic energy, capturing geometric transport costs. Although $\mathcal{E}_g$ lacks explicit dependence on a rate matrix, we demonstrate in Sec. 3.1 that it is equivalent to both $\mathcal{E}_k$ and $\mathcal{E}_c$ under MDM structure.

Finally, since mask schedule $\alpha_t$ determines both the rate matrix and the probability flow in MDM, we reparameterize the energy functionals as $\mathcal{E}_k(\alpha_t, \gamma_t)$, $\mathcal{E}_c(\alpha_t, \gamma_t)$, and $\mathcal{E}_g(\alpha_t, \gamma_t)$ for the rest of paper.

## 3 Main Results

### 3.1 Equivalence of Energies in MDMs

We first formally establish that the three energy functionals defined in Sec. 2the kinetic energy $\mathcal{E}_k$, the conditional kinetic energy $\mathcal{E}_c$, and the geodesic energy $\mathcal{E}_g$are mathematically equivalent under the MDM framework. This equivalence is captured by the following theorems.

**Theorem 3.1** (Kinetic-conditional equivalence in MDMs). *For any weight function $\gamma_t$ and MDM with mask schedule $\alpha_t$, the marginal and conditional kinetic energies are proportional:*

$$\mathcal{E}_k(\alpha_t, \gamma_t) = C_1 \mathcal{E}_c(\alpha_t, \gamma_t), \tag{10}$$

*where $C_1$ is a scalar depending only on the sequence length $n$ and vocabulary size $d$. As a result, the two objectives share the same minimizers:*

$$\arg\min_{\alpha_t} \mathcal{E}_k(\alpha_t, \gamma_t) = \arg\min_{\alpha_t} \mathcal{E}_c(\alpha_t, \gamma_t). \tag{11}$$

As discussed in Sec. 2, while $\mathcal{E}_k$ and $\mathcal{E}_c$ share a similar structure, their inherent differences between marginal and conditional formulations typically lead to divergent values. However, our proof in Appendix D.1 reveals that by decomposing the concrete score in the rate matrix of MDM (see Eq. (2)) into temporal components and clean-data conditional probabilities [44] and leveraging the inherent simple closed-form of MDM's conditional rate matrix (characterized in Appendix C.3), equivalence between $\mathcal{E}_k$ and $\mathcal{E}_c$ are established, even in high-dimensional regimes.

Remarkbly, Theorem 3.1 establishes $\mathcal{E}_c$ as a theoretically sound surrogate for $\mathcal{E}_k$ in MDMs. Crucially, while $\mathcal{E}_k$ suffers from intractability due to the absence of closed-form $p_t(z)$, $\mathcal{E}_c$ remains

computationally tractable in MDMs since both the conditional flow and rate matrix admit closed-form expressions. Furthermore, unlike $\mathcal{E}_g$, $\mathcal{E}_c$ decomposes along sequence dimensions (see Appendix C.4), enabling per-token analysis and creating a natural bridge to $\mathcal{E}_g$ – an energy inherently defined through per-token conditional probability flows. This connection leads to our following theorem.

**Theorem 3.2** (Conditional-geodesic equivalence in MDMs). *For any weight function $\gamma_t$ and MDM with mask schedule $\alpha_t$, the conditional and geodesic energies are proportional:*

$$\mathcal{E}_c(\alpha_t, \gamma_t) = C_2 \mathcal{E}_g(\alpha_t, \gamma_t), \tag{12}$$

*where $C_2$ is a scalar depending only on the sequence length $n$. This implies that they share the same minimizers:*

$$\arg\min_{\alpha_t} \mathcal{E}_c(\alpha_t, \gamma_t) = \arg\min_{\alpha_t} \mathcal{E}_g(\alpha_t, \gamma_t). \tag{13}$$

Our proof in Appendix D.2 reveals that this equivalence originates from the token-wise decomposition of $\mathcal{E}_c$ (see Appendix C.4) – a property inherently enabled by MDM's per-token structure of $p_{t|1}$ and $Q_{t|1}$. This extends the DFM framework [37] beyond its original one-dimensional geodesic-kinetic energy correspondence. Moreover, more flexible $\gamma_t$ choices are considered in our settings while energy functionals in [37] use a fixed weight function $\gamma_t = t$. Therefore, our Theorem 3.2 establishes interpretations applicable to real-world text generation with adaptive weighting schemes.

The insight that the sampling-oriented kinetic energy and the geometrically-motivated geodesic energy constitute two equivalent viewpoints suggests MDMs may simultaneously achieve optimality in both probability flow and rate matrix characteristics despite its structural constraints, which we formally prove in Sec. 3.2. To exemplify these equivalences, we consider the single-token case ($n = 1$) in Example 3.3, where all three energy functionals collapse to an identical closed-form expression. See Appendix D.3 for the proof.

**Example 3.3.** *When $n = 1$, the kinetic, conditional kinetic, and geodesic energies all reduce to:*

$$\mathcal{E}(\alpha_t, \gamma_t) = \int_0^1 \frac{1}{\dot{\gamma}_t} \cdot \frac{\dot{\alpha}_t^2}{\alpha_t(1 - \alpha_t)} \, dt. \tag{14}$$

## 3.2 MDMs with the Optimal Mask Schedule Are Energy Minimizers

Building on the energy equivalence established in Sec. 3.1, we now turn to the core optimization question: *Can an appropriately chosen MDM schedule simultaneously minimize the primary objective kinetic energy $\mathcal{E}_k$ or, by equivalence, all three energy functionals?*

To resolve this question, we initiate our analysis from the geodesic perspective. As discussed in Sec. 2, $\alpha_t$ governs the unmasking process in MDMs, while $\gamma_t$ parametrizes the corresponding continuous interpolation along geodesic curves [42] and also serves as the weight function in the geodesic energy. This schedule duality $\alpha_t$ for discrete dynamics, $\gamma_t$ for geometric flow naturally necessitates an optimal parametric relationship between them, as formalized in the following condition.

**Condition 3.4** (Optimal scheduling condition). *We say that the optimal scheduling condition between the mask schedule $\alpha_t^\star$ and the interpolation schedule $\gamma_t$ is satisfied when*

$$\alpha_t^\star = \sin^2\left(\frac{\pi}{2}\gamma_t\right). \tag{15}$$

The monotonic bijection in the condition, namely $f : [0, 1] \to [0, 1], \quad x \mapsto \sin^2\left(\frac{\pi}{2}x\right)$, creates an *one-to-one schedule correspondence* between $\alpha_t$ and $\gamma_t$.

While this relationship was geometrically established in [42], demonstrating the relationship between the interpolation schedule of the geodesic curve and the mask schedule of MDM that generates the curve, its profound implications for energy minimization remain uncharacterized – a gap our subsequent lemma addresses.

**Lemma 3.5** (Geodesic energy minimization). *Under Condition 3.4, the schedule $\alpha_t^\star$ minimizes the geodesic energy.*

Our proof in Appendix D.4 demonstrates through energy-theoretic analysis that Condition 3.4 not only guarantees generation of minimal-length geodesic paths (thereby providing a proof for existing

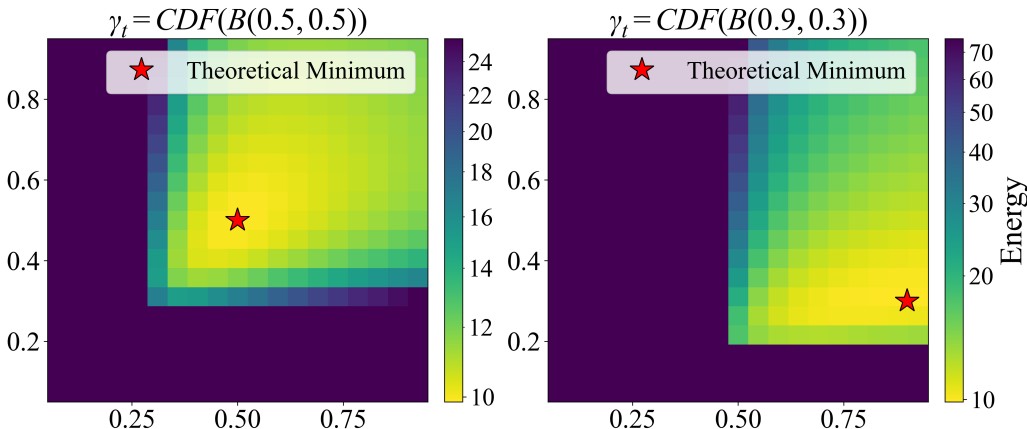

Figure 2: **Distinct weight functions $\gamma_t$ shape different energy landscapes and consequently yield different optimal mask schedules $\alpha_t^\star$.** Axes represent the beta-parameterization of $\alpha_t$ (see Sec. 3.3). Color intensity indicates energy values from Eq. (14). Red stars mark the theoretical minima under the optimal schedule condition.

geometric conclusions [42] in an alternative perspective) but also formally establishes the attainment of minimal geodesic energy. These results enrich our understanding of MDM optimality. One may further be curious about the practical implications: does discretizing the continuous trajectory into a finite-step sampling process affect the attainment of optimality? We address this in Appendix D.5, demonstrating that the discretized trajectory remains optimal in a well-defined sense. Therefore, our subsequent discussion will continue to focus on the theoretical continuous case.

This energy minimization perspective provides a crucial theoretical bridge connecting MDMs' geometric properties with their sampling dynamics. Combining Lemma 3.5 with our equivalence theorems in Sec. 3.1, we extend the minimization result to $\mathcal{E}_k$ and $\mathcal{E}_c$ – energy functionals of primary practical interest due to their direct connection to MDM sampling efficiency, arriving at the following central result under identical optimal scheduling conditions.

**Theorem 3.6** (Kinetic energy minimization). *Under Condition 3.4, the MDM schedule $\alpha_t^\star$ simultaneously minimizes all three energy functionals.*

Theorem 3.6 does not trivially follow from Lemma 3.5, as $\mathcal{E}_k$ and $\mathcal{E}_c$ require simultaneous optimization of probability flows and rate matrices – a fundamental departure from $\mathcal{E}_g$'s exclusive dependence on the probability flow. Our proof in Appendix D.6 crucially relies on the fact that although the mask schedule $\alpha_t$ governs $p_t$ and $Q_t$ jointly in MDMs, introducing parametric constraints, the Markov structure of MDM still intrinsically co-optimizes the probability flow and the rate matrix. This resolves a long-standing conceptual paradox in discrete diffusion: MDM's simple coupled framework in fact preserves optimal transport properties through its intrinsic design of Markovian transitions.

Furthermore, Theorem 3.6 establishes that Condition 3.4 not only dictates optimal probability paths but also governs optimal sampling rates. This theoretical insight directly motivated our energy-inspired schedule tuning method in Sec. 3.3, where we select $\alpha_t^\star$ that minimizes energies given fixed $\gamma_t$. To empirically validate how distinct weight functions $\gamma_t$ shape different energy landscapes and consequently yield unique optimal mask schedules, we revisit the single-token case ($n = 1$) in Example 3.3. Fig. 2 quantitatively demonstrates this relationship by visualizing how varying $\gamma_t$ induces corresponding $\alpha_t^\star$ schedules that minimize the energy functional defined in Eq. (14).

## 3.3 Energy-Inspired Fast Samplers

Our energy minimization perspective introduced in Sec. 3.2 shows the importance of the weight function $\gamma_t$. Especially, the term $(\dot{\gamma}_t)^{-1}$ in the energy functionals (See Definition 2.1, 2.2 and 2.3) plays a central role: it downweights regions of rapid temporal change in $\gamma_t$, effectively focusing optimization on slower regions. Different choices of $\gamma_t$ thus encode different emphases in the diffusion processfor instance, whether to spend more computational budget early (coarse structure) or late (fine detail) in the unmasking trajectory. This observation suggests that task-specific tuning

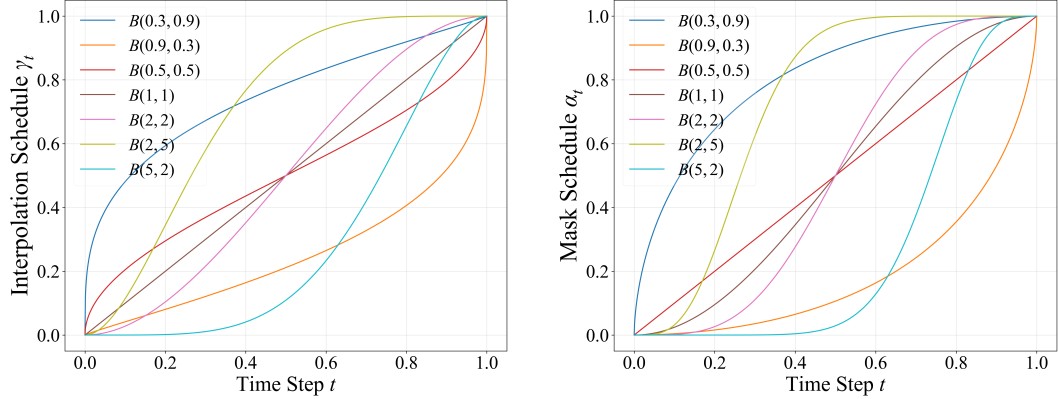

Figure 3: **Beta-parameterized interpolation schedules and corresponding mask schedules.** The left panel demonstrates beta-parameterized interpolation schedule morphologies, while the right panel displays corresponding optimal $\alpha_t^\star$ schedules derived via Condition 3.4.

of $\gamma_t$ may yield performance improvements, as different tasks may benefit from different temporal allocations.

Moreover, since MDM training objectives are invariant to the choice of schedule $\alpha_t$ (as discussed in Sec. 2.1), schedule optimization can be performed post hocafter model trainingwithout requiring retraining. This enables lightweight adaptation of pretrained models to new distributions or generation objectives by simply modifying the sampling schedule.

However, direct optimization over the space of all possible schedules remains intractable due to its infinite-dimensional nature. Existing approaches therefore rely on a small set of manually designed schedules, such as linear ($\alpha_t = t$) [8, 26, 4], sine ($\alpha_t = \sin\left(\frac{\pi}{2}t\right)$) [36], or squared sine schedules ($\alpha_t = \sin^2\left(\frac{\pi}{2}t\right)$) [17]. To bridge the gap between theoretical flexibility and practical feasibility, we propose parameterizing $\gamma_t$ as the cumulative distribution function (CDF) of a beta distribution with two parameters $(a, b)$:

$$\gamma_t = \text{CDF}_{\mathcal{B}(a,b)}(t), \tag{16}$$

which, via the condition in Eq. (15), yields a corresponding $\alpha_t$ schedule. This parametric framework generates diverse schedule topologies including convexity and inflection points through just two parameters (see Fig. 3) and it is motivated by a simple observation formalized in the following proposition, proved in Appendix D.7.

**Proposition 3.7.** *Linear and squared sine schedules correspond to specific beta parameterizations:*

$$\alpha_t = t \quad \Leftrightarrow \quad \gamma_t = CDF_{\mathcal{B}(0.5,0.5)}(t), \tag{17}$$

$$\alpha_t = \sin^2\left(\frac{\pi}{2}t\right) \quad \Leftrightarrow \quad \gamma_t = t = CDF_{\mathcal{B}(1,1)}(t). \tag{18}$$

We begin with a toy model in Fig. 4 to illustrate how tuning beta parameters $(a, b)$ affects sampling quality under low-step regimes, showing that different target distributions prefer different schedules, highlighting the need for task-specific adaptation. In Section 4, we further demonstrate that, certain beta schedules can outperform standard hand-crafted schedules, especially when the number of sampling steps is limited, on real benchmarks.

## 4 Experiments

In this section, we demonstrate that our energy-inspired task-specific tuning method introduced in Sec. 3.3 can be used to accelerate MDM sampling in practical applications such as mathematical reasoning and code generation. Crucially, the invariance of training loss of MDMs to the choice of mask schedules (see Section 2.1) allows us to efficiently optimize the mask schedule for specific downstream tasks without the computational burden of end-to-end model retraining. For details on how to identify a task-favorable schedule by tuning the beta parameters, please refer to Appendix E.1.

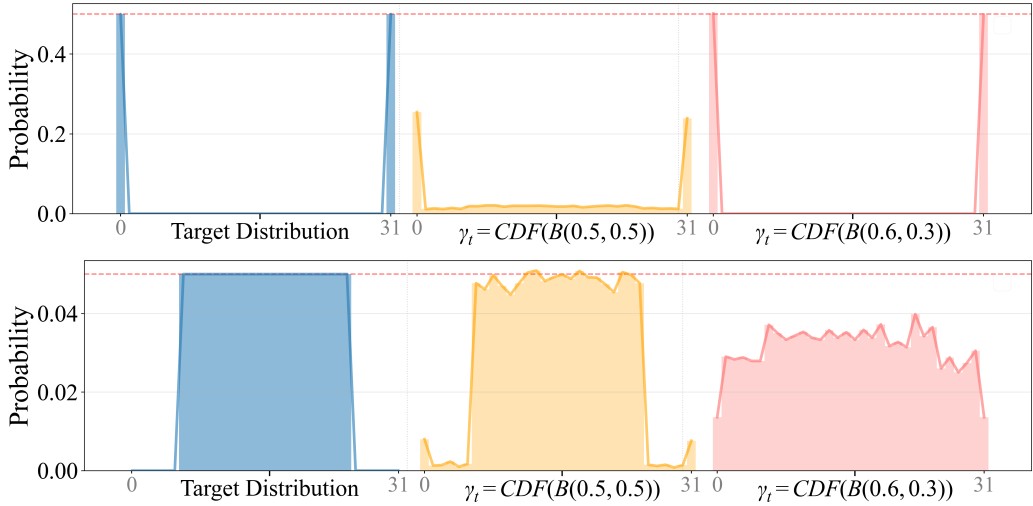

Figure 4: **Toy experiments illustrating how different target distributions prefer different schedules.** Each panel visualizes the effect of beta parameter tuning on sampling quality under limited step budgets by showing a target distribution and two distributions sampled by different schedules. More details of this experiment are provided in Appendix E.6.

We evaluate our method using LLaDA 8B [43], an open-source MDM that achieves performance comparable to modern large language models such as LLaMA3 [31]. We select six representative tasks: MBPP [7], HumanEval [9], BBH [28], GSM8K [10], Hendrycks Math [11] and Minerva Math [18]. These benchmarks comprehensively assess the models capabilities in general reasoning, mathematical problem solving, and code generation. Please refer to Appendix E.2 for more experimental details.

Fig. 5 systematically evaluates sampling performance across diverse reasoning benchmarks under varying step budgets. Our analysis reveals that on code generation tasks (MBPP and HumanEval), beta-parameterized schedules match the generation quality of the linear baseline with $2\times$ step reduction. For the Hendrycks Math mathematical reasoning task, our method achieves performance parity with the linear schedule using $4\times$ fewer steps. There are also benchmarks on which beta-parameterized schedules exhibit comparable yet not better performance, such as BBH [28] and GSM8K [10], and we provide the results on these two benchmarks in Appendix. E.3.

On benchmarks where beta-parameterized schedules demonstrate profound empirical advantages over manual baselines, we observe a systematic preference for convex interpolation schedules. As analyzed in Sec. 3.3, this empirical bias suggests that for certain tasks such as code generation, allocating computational resources to optimize early-stage sampling dynamics (coarse structure formation) may yield greater quality gains compared to fine-grained refinement phases. Further discussions on the task-specific schedule preferences are presented in Appendix E.4.

While more rigorous characterization of task-schedule correspondences remains open, constituting critical directions for future research, the schedule invariance property facilitates computationally efficient schedule exploration without model retraining. Practitioners can thus perform task-specific schedule tuning through our framework, requiring no additional training infrastructure.

Raw data of our experiments corresponding to Fig. 5 are presented in Appendix E.3, and we provide additional samples in Appendix E.5 to offer a more comprehensive understanding.

## 5   Related Work

**Mask Diffusion Models**. MDMs [4, 8, 34, 44, 36, 38] have established themselves as a prominent class of generative models for discrete data. While substantial progress has been made in understanding MDM training dynamics through theoretically equivalent objective formulations [12, 38,

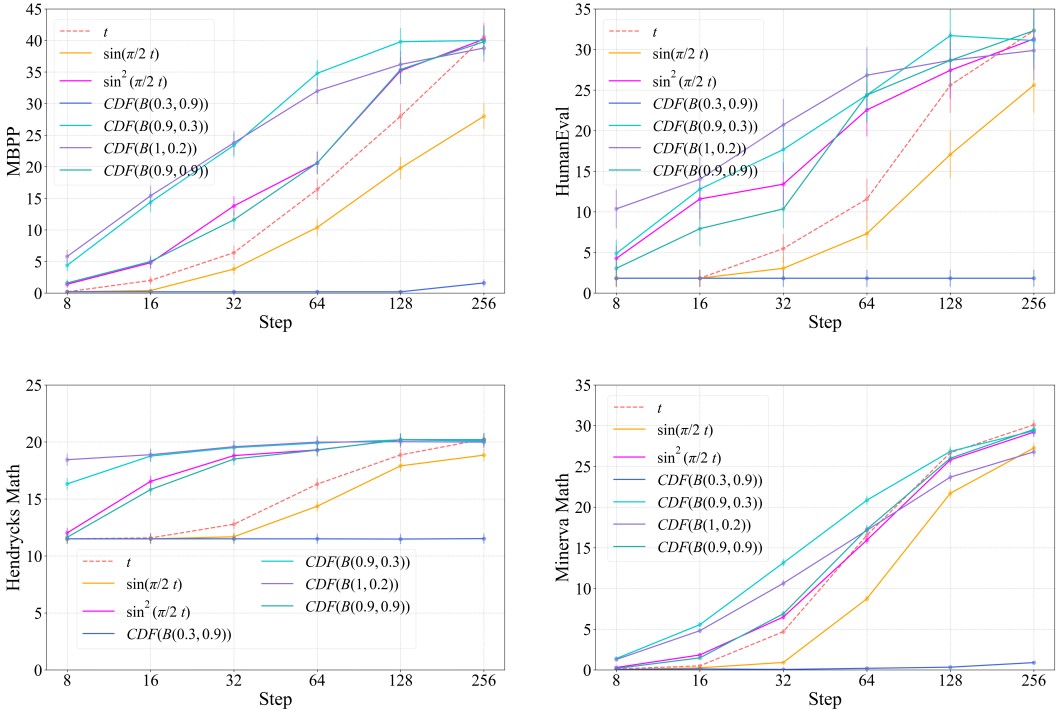

Figure 5: **Performance evaluation of energy-optimized schedules on LLaDA 8B [43].** Each panel corresponds to a distinct benchmark. The x-axis displays sampling steps on a logarithmic scale, while the y-axis quantifies task performance, where higher values denote superior generation quality. Results on benchmarks where beta-parameterized schedules exhibit comparable yet not better performance are provided in Appendix. E.3.

36, 44] and parameterization strategies [8, 14, 21, 20, 19], the sampling process remains relatively underexplored. Existing efforts primarily concentrate on developing advanced discrete sampling algorithms, including Tweedie $\tau$-sampling [3, 6, 26], k-Gillespie methods [48], and higher-order solvers [45], along with discretizing time step optimization techniques [35] and distillation-based acceleration [33, 46]. Notably, the critical mask schedules governing sampling trajectories have not received systematic investigation. Current implementations [8, 26, 4, 17, 36] typically employ manually designed schedules (predominantly linear), even in state-of-the-art MDM-based large language models [43]. This underscores the necessity for principled task-adaptive schedule optimization, which constitutes our primary contribution.

**Energy and Geodesic Perspective**. The connection between kinetic energy minimization [2, 13] and efficient sampling via optimal transport trajectories [24, 22, 25, 29] has been well-established in continuous settings. However, existing literature predominantly examines continuous diffusion processes, limiting direct applicability to discrete domains. Recent geometric analyses [42] reveal intrinsic links between MDM probability paths and geodesic curves under specific interpolation schedules, motivating our exploration of geodesic perspective as a bridge between MDMs and kinetic principles. However, the energy perspectivewhich plays a vital role in our optimal transport frameworkis absent in [42], thus preventing it from establishing the optimality of Condition 3.4 in the context of sampling efficiency. The most relevant work [37] introduces energy perspectives to Discrete Flow Matching (DFM), a distinct discrete probability flow (DPF) variant. Crucially, their framework decouples probability flow and rate matrix optimization - an approach incompatible with MDMs where $\alpha_t$ jointly governs both components. Despite this architectural constraint, we demonstrate through Theorem 3.6 that MDMs inherently achieve optimal rate selection by leveraging a key lemma from [37] (see Appendix D.6). Our introduced $\gamma_t$ interpolation schedule also provides novel theoretical insights by establishing that every mask schedule optimizes a corresponding energy functional, thereby justifying task-specific schedule tuning - a capability absent in [37] where formulations reduce to the $\gamma_t = t$ special case. Furthermore, the training loss invariance to mask

schedules represents a unique MDM property distinguishing it from DFM and conventional DPF frameworks, enabling exclusive post-training schedule optimization within our MDM paradigm.

# 6 Conclusion

We present a theoretical framework that establishes MDMs as optimal transport processes minimizing three distinct energy formulations, demonstrating that MDMs inherently achieve minimal sampling cost through energy-optimal mask schedules. Building upon this theoretical foundation, we develop a Beta-CDF parameterization scheme that facilitates efficient task-adaptive schedule optimization. Comprehensive empirical validation across synthetic and real-world benchmarks confirms our framework's effectiveness, showing consistent performance gains in few-step sampling scenarios.

**Limitation.** While our method enables practical task-specific schedule optimization, the intrinsic relationship between high-dimensional real-world tasks and their optimal schedules remains not fully understood - a fundamental challenge requiring further investigation. This limitation highlights promising directions for future research in interpretable schedule-task correlation analysis.

**Broader Impact.** Our schedule tuning framework could benefit real-world applications by reducing computational costs. Conversely, the acceleration capability may lower synthetic content generation barriers, potentially exacerbating misinformation risks.

# Acknowledgement

This work was supported by the National Natural Science Foundation of China (No. 92470118); the Beijing Natural Science Foundation (No. L247030); the Beijing Nova Program (No. 20230484416); the Public Computing Cloud of Renmin University of China; the Beijing Major Science and Technology Project under Contract no. Z251100008425002; the fund for building world-class universities (disciplines) of Renmin University of China; and the Huawei Research Fund.

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

# Contents

# Appendix A  Detailed Notations and Definitions

- $n$: the sequence length.
- $x, z$: a $n$-dimensional vector representing states in a model.
- $\mathcal{D}$: the vocabulary with size $|\mathcal{D}| = d$.
- $x^i, z^i \in \mathcal{D}$: the $i$-th token of data $x, z$.
- $m(z)$: the number of mask tokens in $z$.
- $(p_t(z))_{t \in [0,1]}$: the DPF that connects a simple distribution $p_0(z)$ and a data distribution $p_1(z) = q(z)$.
- $p_{t|1}(z|x_1)$: the conditional probability flow conditioned on the data.
- $p_{1|t}(x_1|z)$: the posterior distribution conditioned on time $t$.
- $\alpha_t$: The mask schedule function.
- $Q_t$: The transition rate matrix of the unmasking process of MDM at time $t$.
- $Q_{t|1}$: The conditional rate matrix of the unmasking process of MDM at time $t$.
- $\overleftarrow{Q}_t$: The transition rate matrix of the masking process of MDM at time $t$.
- $\sigma_t$: The transition rate that uniquely determines $\overleftarrow{Q}_t$ in MDM settings.
- $\gamma_t$: The interpolation schedule function of the geodesic curve on the high-dimensional sphere. Also the weight function we choose in three energy functionals for energy minimization.
- $B(a, b)$: Beta distribution with parameters $a$ and $b$.
- CDF: cumulative distribution function

# Appendix B  An Intuitive Explanation of Geodesics

This appendix provides an intuitive introduction to geodesics and exponential maps to clarify the geometric interpretation of MDM in [42]. Readers seeking formal mathematical definitions of these differential geometry concepts may refer to [5] for complete technical specifications.

**Manifolds and Tangent Spaces**. A manifold is a smooth high-dimensional surface, such as a $\mathcal{D}$-dimensional sphere $\mathbb{S}^{D-1}$. In our scenario, the embedding Eq. (8) maps the per-token conditional probability flow Eq. (3) onto $\mathbb{S}^{D-1}$, as shown in Sec. 2.3. On every point $y_0$ on the manifold, there exists a tangent space $\mathcal{T}_{y_0}$ containing all vectors starting from $y_0$ and tangent to the manifold.

**Exponential Map and Geodesics**. The exponential map is denoted as $\exp_{y_0}(v)$, which maps a tangent vector $v \in \mathcal{T}_{y_0}$ to a point $y_1$ on the manifold. Geometrically, this represents moving from $y_0$ along the "direction" of $v$ at constant speed until reaching $y_1$. This movement follows a geodesic path, which is both the "shortest path" and the "straight path" between two points on a manifold, generalizing straight lines in Euclidean space. For example, great circles are geodesics on spheres.

**Inverse Exponential Map and Parameterized Geodesic Trajectory**: The inverse exponential map is denoted as $\exp_{y_0}^{-1}(y_1)$, which maps a manifold point $y_1$ back to a tangent vector $v \in \mathcal{T}_{y_0}$. This vector encodes both direction and distance from $y_0$ to $y_1$. Therefore, given start/end points $y_0$ and $y_1$ on the manifold, a geodesic trajectory parameterized by $\gamma_t$ (strictly increasing with $\gamma_0 = 0$, $\gamma_1 = 1$) can be expressed as:

$$\exp_{y_0}^{-1}(y_t) = \gamma_t \cdot \exp_{y_0}^{-1}(y_1), \quad t \in [0, 1] \tag{19}$$

This formulation implies:

- $\exp_{y_0}^{-1}(y_1)$ is the tangent vector encoding the direction and scale that generates the geodesic curve from $y_0$ to $y_1$.
- $\gamma_t$ is the interpolation schedule and $\gamma_t = t$ means constant-speed motion along the geodesic.

Recent theoretical advances [42] reveal that MDM's conditional probability flow in Eq. (3) forms exactly the geodesic curve in spherical geometry (see Fig. 6). The interpolation schedule $\gamma_t$ is uniquely determined by the mask schedule $\alpha_t$ as $\gamma_t = \frac{2}{\pi} \arcsin \sqrt{\alpha_t}$, which is equivalent with Condition 3.4.

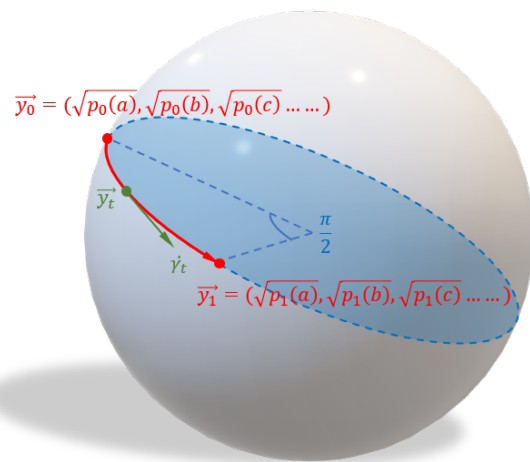

Figure 6: **The per-token conditional probability flow in MDM generates exactly the geodesic curve.**

## Appendix C  Proof of Auxillary Lemmas

In this section, we provide complete proofs for the auxiliary lemmas referenced in the main text for completeness.

### C.1  Derivation of the Relationship between $\alpha_t$ and $\sigma_t$ in MDM

**Lemma C.1.** *The mask schedule $\alpha_t$ relates to the rate $\sigma_t$ via expression*

$$\alpha_t = \exp\left(-\int_t^1 \sigma_s ds\right). \tag{20}$$

*Proof.* From the definition of mask schedule in Eq. (3), it suffices to prove

$$\mathbb{P}(x_t^i = x_1^i | x_1) = \exp\left(-\int_t^1 \sigma_s ds\right). \tag{21}$$

Consider infinitesimal time intervals $(t, t - \Delta t]$ where each token experiences masking probability $\sigma_t \Delta t + o(\Delta t)$. The preservation probability therefore satisfies the following product bounds:

$$\prod_{k=0}^{\lfloor(1-t)/\Delta t\rfloor+1}(1 - \sigma_{(1-k\Delta t)}\Delta t + o(\Delta t)) \leq \mathbb{P}(x_t^i = x_1^i | x_1) \tag{22}$$

$$\leq \prod_{k=0}^{\lfloor(1-t)/\Delta t\rfloor}(1 - \sigma_{(1-k\Delta t)}\Delta t + o(\Delta t)). \tag{23}$$

Analyzing the upper bound through logarithmic transformation, we get

$$\prod_{k=0}^{\lfloor(1-t)/\Delta t\rfloor}(1 - \sigma_{(1-k\Delta t)}\Delta t + o(\Delta t)) = \exp\left(\sum_{k=0}^{\lfloor(1-t)/\Delta_t\rfloor}\log(1 - \sigma_{1-k\Delta t}\Delta t + o(\Delta t))\right) \tag{24}$$

$$= \exp\left(\sum_{k=0}^{\lfloor(1-t)/\Delta_t\rfloor}(-\sigma_{1-k\Delta t}\Delta t + o(\Delta t))\right) \tag{25}$$

$$\overset{(1)}{\to} \exp\left(-\int_0^{1-t}\sigma_{1-u}du\right) \tag{26}$$

$$\overset{(2)}{=} \exp\left(-\int_t^1\sigma_s ds\right), \tag{27}$$

where in (1) follows from Riemann sum convergence as $\Delta t \to 0$ and (2) applies the variable substitution $s = 1 - u$ to align integration limits.

The lower bound converges identically through analogous arguments. This completes the proof. $\square$

## C.2 Proof of the Invariance of Training Loss to the Mask Schedule

Different equivalence expressions of the training loss of MDM has been proved invariant to the choice of $\alpha_t$ in multiple works [12, 38, 36]. We adapt a proof from [36] by examining the negative evidence lower bound (NELBO) through token-level denoising components:

$$\mathcal{L}_{\text{NELBO}} = \mathbb{E}_{p_1(x_1),p_{t|1}(z|x_1)} \int_0^1 \frac{-\dot{\alpha}_t}{1 - \alpha_t} \log\langle x_\theta(z_t, t), x_1 \rangle \mathrm{d}t \tag{28}$$

$$= \mathbb{E}_{p_1(x_1),p_{t|1}(z|x_1)} \int_0^1 \frac{-\dot{\alpha}_t}{1 - \alpha_t} \sum_{i=1}^n \log\langle x_\theta^i(z^{1:n}, t), x_1^i \rangle \mathrm{d}t. \tag{29}$$

Despite the apparent dependence on $\alpha_t$ in its parametric form, the loss exhibits fundamental invariance as formalized below.

**Proposition C.2** (Schedule Invariance, Proof Adapted from [36]). *$\mathcal{L}_{NELBO}$ is invariant to $\alpha_t$'s functional form, depending only on its boundary values $\alpha_0 = 0, \alpha_1 = 1$.*

*Proof.* The invariance emerges through variable substitution via the chain rule. Let $\gamma \equiv \log(1 - \alpha_t)$, then

$$\mathcal{L}_{\text{NELBO}} = \mathbb{E}_{p_1(x_1),p_{t|1}(z|x_1)} \int_{t=0}^{t=1} \frac{-\alpha_t'}{1 - \alpha_t} \log\langle x_\theta(z_t, t), x \rangle \mathrm{d}t \tag{30}$$

$$\overset{(1)}{=} \mathbb{E}_{p_1(x_1),p_{t|1}(z|x_1)} \int_{t=0}^{t=1} \log\langle x_\theta(z_t, t), x \rangle \mathrm{d}[f(t)] \tag{31}$$

$$\overset{(2)}{=} \mathbb{E}_{p_1(x_1),p_{t|1}(z|x_1)} \int_{\gamma=0}^{\gamma=-\infty} \log\langle x_\theta(z_{f^{-1}(\gamma)}, f^{-1}(\gamma)), x \rangle \mathrm{d}\gamma \tag{32}$$

$$\overset{(3)}{=} -\mathbb{E}_{p_1(x_1),p_{t|1}(z|x_1)} \int_{\gamma=-\infty}^{\gamma=0} \log\langle \tilde{x}_\theta(\tilde{z}_\gamma, \gamma), x \rangle \mathrm{d}\gamma \tag{33}$$

Here (1) applies the substitution $f(t) = \log(1 - \alpha_t)$. (2) applies change of variable $\gamma \equiv f(t)$. In (3) we let $\tilde{z}_\gamma \equiv z_{f^{-1}(\gamma)}$, $\tilde{x}_\theta(\tilde{z}_\gamma, \gamma) \equiv x_\theta(\tilde{z}_\gamma, f^{-1}(\gamma))$. The final expression contains no explicit dependence on $\alpha_t$'s trajectory between its fixed endpoints, thereby establishing the claimed invariance. $\square$

## C.3 Derivation of Conditional Rate Matrix of MDM

Although explicit sampling through $Q_{t|1}(z, x|x_1)$ remains unnecessary in MDM, establishing closed-form representations proves valuable for theoretical characterization.

**Lemma C.3** (Conditional Rate of MDM). *The following conditional rate matrix generates MDM's unmasking process:*

$$Q_t(z, x|x_1) = \begin{cases} \overleftarrow{Q_t}(x, z) \frac{p_{t|1}(x|x_1)}{p_{t|1}(z|x_1)} & p_{1|t}(x_1|z) > 0 \\ 0 & \textit{otherwise} \end{cases} = \begin{cases} \frac{\sigma_t \alpha_t}{1 - \alpha_t} & z \to x \Rightarrow x_1 \\ -\sum_{x \neq z} Q_t(z, x|x_1) & x = z \\ 0 & \textit{otherwise} \end{cases}. \tag{34}$$

*Here $z \to x$ denotes single-token unmasking transitions defined in Sec.2.1 and $x \Rightarrow x_1$ denotes that $x_1$ can be generated from $x$ through one or several steps of unmasking.*

Applying Bayes' theorem establishes the posterior relationship:

$$p_{1|t}(x_1|z) = \frac{p_{t|1}(z|x_1)p_1(x_1)}{p_t(z)}, \tag{35}$$

we know that the positivity condition $p_{1|t}(x_1|z) > 0$ consequently requires $p_{t|1}(z|x_1) > 0$ and $z \Rightarrow x_1$, ensuring the rate matrix's well-posedness.

*Proof.* To verify that $Q_t(z, x|x_1)$ generates MDM's demasking dynamics, we confirm the consistency condition in Eq. (6) holds. We verify this through the following derivation:

$$\sum_{x_1} Q_t(z, x|x_1)p_{1|t}(x_1|z) = \sum_{x_1} Q_t(z, x|x_1)\frac{p_{t|1}(z|x_1)p_1(x_1)}{p_t(z)} \tag{36}$$

$$= \sum_{x_1} \overleftarrow{Q}_t(x, z)\frac{p_{t|1}(x|x_1)}{p_{t|1}(z|x_1)}\frac{p_{t|1}(z|x_1)p_1(x_1)}{p_t(z)} \tag{37}$$

$$= \sum_{x_1} \overleftarrow{Q}_t(x, z)\frac{p_{t|1}(x|x_1)p_1(x_1)}{p_t(z)} \tag{38}$$

$$= \overleftarrow{Q}_t(x, z)\frac{p_t(x)}{p_t(z)} = Q_t(z, x). \tag{39}$$

We subsequently derive its closed-form of $Q_t(z, x|x_1)$. If $x \not\Rightarrow x_1$, then $p_{t|1}(x|x_1) = 0$, yielding $Q_t(z, x|x_1) = 0$. If $z \to x \Rightarrow x_1$, on the other hand, we have

$$Q_t(z, x|x_1) = \overleftarrow{Q}_t(x, z)\frac{p_{t|1}(x^i|x_1^i)}{p_{t|1}(z^i|x_1^i)}$$
$$= \sigma_t\frac{\alpha_t}{1 - \alpha_t}, \tag{40}$$

thus completing the proof. $\qquad\square$

## C.4 Decomposition of Conditional Kinetic Energy along Sequence Dimension

In this section, we show that under many DPF frameworks such as MDMs, the conditional kinetic energy can be decomposed along sequence dimension as following:

$$\mathcal{E}_c(p_{t|1}, Q_{t|1}; \gamma_t) = \mathbb{E}_{t,p_1(x_1),p_{t|1}(z|x_1)} \sum_{x:x\neq z} \frac{1}{\dot{\gamma}_t p_{t|1}(x|x_1)}Q_{t|1}(z, x|x_1)^2 \tag{41}$$

$$= \mathbb{E}_{t,p_1(x_1)} \sum_{z,x:x\neq z} \frac{p_{t|1}(z|x_1)}{\dot{\gamma}_t p_{t|1}(x|x_1)}Q_{t|1}(z, x|x_1)^2 \tag{42}$$

$$\overset{(1)}{=} \mathbb{E}_{t,p_1(x_1)} \sum_{i=1}^{n} C \sum_{z^i,x^i:x^i\neq z^i} \frac{p_{t|1}(z^i|x_1)}{\dot{\gamma}_t p_{t|1}(x^i|x_1)}Q_{t|1}(z^i, x^i|x_1)^2 \tag{43}$$

$$= \sum_{i=1}^{n} C\mathbb{E}_{t,p_1(x_1),p_{t|1}(z^i|x_1)} \sum_{x^i:x^i\neq z^i} \frac{1}{\dot{\gamma}_t p_{t|1}(x^i|x_1)}Q_{t|1}(z^i, x^i|x_1)^2 \tag{44}$$

The pivotal decomposition in (1) leverages two structural properties: (i) The conditional probability flow (Eq. (3)) exhibits token-wise independence, and (ii) The conditional rate matrix (Lemma C.3) nullifies transitions altering multiple tokens simultaneously. These enable reduction of full-sequence transitions to single-token operations, with remaining $n-1$ tokens contributing constant combinatorial factors. This structural property persists across various DPF implementations including MDM and Discrete Flow Matching [32, 37], validating the conditional kinetic energy as theoretically sound surrogate objective. Notably, standard kinetic energy $\mathcal{E}$ lacks such decomposition due to $p_t(z)$'s dependence on cross-token correlations.

For MDM's binary mask dynamics ($x_1^i$ vs. [MASK]), the combinatorial constant becomes $C = 2^{n-1}$:

$$\mathcal{E}_c(\alpha_t, \gamma_t) = 2^{n-1} \cdot \sum_{i=1}^{n} \mathbb{E}_{t,p_1(x_1),p_{t|1}(z^i|x_1)} \sum_{x^i:x^i\neq z^i} \frac{1}{\dot{\gamma}_t p_{t|1}(x^i|x_1)}Q_{t|1}(z^i, x^i|x_1)^2. \tag{45}$$

This decomposition permits notational relaxation where $\mathcal{E}_c$ analysis considers tokens $z, x \in \mathcal{D}$ independently of sequence context, a slight abuse off notation adopted in Appendix D.2's equivalence proof.

## Appendix D   Proof of Main Results

### D.1   Proof of Theorem 3.1

**Theorem 3.1** (Kinetic-conditional equivalence in MDMs). *For any weight function $\gamma_t$ and MDM with mask schedule $\alpha_t$, the marginal and conditional kinetic energies are proportional:*

$$\mathcal{E}_k(\alpha_t, \gamma_t) = C_1 \mathcal{E}_c(\alpha_t, \gamma_t), \tag{46}$$

*where $C_1$ is a scalar depending only on the sequence length $n$ and vocabulary size $d$. As a result, the two objectives share the same minimizers:*

$$\arg\min_{\alpha_t} \mathcal{E}_k(\alpha_t, \gamma_t) = \arg\min_{\alpha_t} \mathcal{E}_c(\alpha_t, \gamma_t). \tag{47}$$

Our argument leverages a foundational decomposition from [44] regarding concrete score representations:

**Lemma D.2.** *for $z = (z^1, ..., z^i = [\mathrm{M}], ...z^n)$, $x = (z^1, ..., x^i \neq [\mathrm{M}], ...z^n)$, we have*

$$\frac{p_t(x)}{p_t(z)} = \frac{\alpha_t}{1 - \alpha_t} p_1(x^i | z^{UM}), \tag{48}$$

*where $z^{UM}$ is the vector consists of all unmasked tokens of $z$.*

We now prove the main theorem:

*Proof.* Let $m(z)$ quantify the masked positions in $z$, assumed w.l.o.g. to occupy initial sequence positions. The key summation decomposes as:

$$\sum_{x:z\to x} \frac{p_t(x)}{p_t(z)} = \sum_{i=1}^{m(z)} \sum_{\substack{x:z\to x \\ x^i \neq z^i = [\mathrm{M}]}} \frac{p_t(x)}{p_t(z)} \tag{49}$$

$$= \sum_{i=1}^{m(z)} \sum_{x^i \neq [\mathrm{M}]} \frac{\alpha_t}{1 - \alpha_t} p_0(x^i | z^{UM}) \tag{50}$$

$$= \sum_{i=1}^{m(z)} \frac{\alpha_t}{1 - \alpha_t} \sum_{x^i \neq [\mathrm{M}]} p_0(x^i | z^{UM}) \tag{51}$$

$$= m(z) \frac{\alpha_t}{1 - \alpha_t}. \tag{52}$$

Substituting the rate matrix from Eq. (2), we therefore deduce

$$\mathcal{E}_k = \mathbb{E}_{t,p_t(z)} \sum_{x:z\to x} \frac{1}{p_t(x)\dot{\gamma}_t} \left( \sigma_t \frac{p_t(x)}{p_t(z)} \right)^2 \tag{53}$$

$$= \mathbb{E}_t \sum_z \sum_{x:z\to x} \frac{p_t(z)}{p_t(x)\dot{\gamma}_t} \sigma_t^2 \frac{p_t^2(x)}{p_t^2(z)} \tag{54}$$

$$= \mathbb{E}_t \sum_z \sum_{x:z\to x} \frac{\sigma_t^2}{\dot{\gamma}_t} \frac{p_t(x)}{p_t(z)} \tag{55}$$

$$\overset{(1)}{=} \mathbb{E}_t \sum_z m(z) \frac{\alpha_t}{1-\alpha_t} \frac{\sigma_t^2}{\dot{\gamma}_t} \tag{56}$$

$$= \left( \sum_z m(z) \right) \mathbb{E}_t \frac{\alpha_t}{1-\alpha_t} \frac{\sigma_t^2}{\dot{\gamma}_t} \tag{57}$$

$$\triangleq C_k \cdot \mathbb{E}_t \frac{\alpha_t}{1-\alpha_t} \frac{\sigma_t^2}{\dot{\gamma}_t}. \tag{58}$$

Here we define $C_k = \sum_z m(z)$. On the other hand, the conditional kinetic energy can also be break down as

$$\mathcal{E}_c = \mathbb{E}_{t,p_1(x_1),p_{t|1}(z|x_1)} \sum_{x:z\to x} \frac{1}{p_{t|1}(x|x_1)\dot{\gamma}_t} \left( \sigma_t \frac{p_{t|1}(x|x_1)}{p_{t|1}(z|x_1)} \right)^2 \tag{59}$$

$$= \mathbb{E}_{t,p_1(x_1)} \sum_{z,x:z\to x\Rightarrow x_1} \frac{\sigma_t^2}{\dot{\gamma}_t} \frac{p_{t|1}(x|x_1)}{p_{t|1}(z|x_1)} \tag{60}$$

$$= \mathbb{E}_{t,p_1(x_1)} \sum_{z,x:z\to x\Rightarrow x_1} \frac{\alpha_t}{1-\alpha_t} \frac{\sigma_t^2}{\dot{\gamma}_t} \tag{61}$$

$$= \left( \sum_{x_1} \sum_{z,x:z\to x\Rightarrow x_1} p_1(x_1) \right) \mathbb{E}_t \frac{\alpha_t}{1-\alpha_t} \frac{\sigma_t^2}{\dot{\gamma}_t} \tag{62}$$

$$= n2^{n-1} \left( \sum_{x_1} p_1(x_1) \right) \mathbb{E}_t \frac{\alpha_t}{1-\alpha_t} \frac{\sigma_t^2}{\dot{\gamma}_t} \tag{63}$$

$$= n2^{n-1} \mathbb{E}_t \frac{\alpha_t}{1-\alpha_t} \frac{\sigma_t^2}{\dot{\gamma}_t} \tag{64}$$

$$\triangleq C_c \cdot \mathbb{E}_t \frac{\alpha_t}{1-\alpha_t} \frac{\sigma_t^2}{\dot{\gamma}_t}. \tag{65}$$

This expression of $\mathcal{E}_c$ is equivalent to the decomposition in Appendix C.4 by plugging in the explicit form of $Q_{t|1}$. The proportionality constant $C_1 = C_k/C_c$ emerges from comparing both expressions, with $C_k, C_c$ depending solely on architectural parameters $n$ and $|\mathcal{D}| = d$. The minimizer equivalence follows directly from the strict positivity of scaling constants.

$$\square$$

## D.2 Proof of Theorem 3.2

**Theorem 3.2** (Conditional-geodesic equivalence in MDMs). *For any weight function $\gamma_t$ and MDM with mask schedule $\alpha_t$, the conditional and geodesic energies are proportional:*

$$\mathcal{E}_c(\alpha_t, \gamma_t) = C_2 \mathcal{E}_g(\alpha_t, \gamma_t), \tag{66}$$

*where $C_2$ is a scalar depending only on the sequence length $n$. This implies that they share the same minimizers:*

$$\arg\min_{\alpha_t} \mathcal{E}_c(\alpha_t, \gamma_t) = \arg\min_{\alpha_t} \mathcal{E}_g(\alpha_t, \gamma_t). \tag{67}$$

*Proof.* The geodesic energy is inherently defined using the token-wise independent conditional flows in Eq. (3), therefore it admits straightforward decomposition along sequence dimensions. In MDM case where all tokens follows the same mask schedule $\alpha_t$, we further have

$$\mathcal{E}_g(p_{t|1}; \gamma_t) = \sum_{i=1}^{n} \mathbb{E}_{t, p_1(x_1), p_{t|1}(z^i|x_1)} \frac{4}{\dot{\gamma}_t p_{t|1}(z^i|x_1)} \dot{y}_{t|1}(z^i|x_1)^2 \tag{68}$$

$$= n \cdot \mathbb{E}_{t, p_1(x_1), p_{t|1}(z^i|x_1)} \frac{4}{\dot{\gamma}_t p_{t|1}(z^i|x_1)} \dot{y}_{t|1}(z^i|x_1)^2 \tag{69}$$

$$\triangleq C_g \cdot \mathbb{E}_{t, p_1(x_1), p_{t|1}(z^i|x_1)} \frac{4}{\dot{\gamma}_t p_{t|1}(z^i|x_1)} \dot{y}_{t|1}(z^i|x_1)^2 \tag{70}$$

On the other hand, through dimensional decomposition established in Appendix C.4, the conditional energy admits:

$$\mathcal{E}_c(p_{t|1}, Q_{t|1}; \gamma_t) = 2^{n-1} \cdot \sum_{i=1}^{n} \mathbb{E}_{t, p_1(x_1), p_{t|1}(z^i|x_1)} \sum_{x^i : x^i \neq z^i} \frac{1}{\dot{\gamma}_t p_{t|1}(x^i|x_1)} Q_{t|1}(z^i, x^i|x_1)^2 \tag{71}$$

$$= C_c \cdot \mathbb{E}_{t, p_1(x_1), p_{t|1}(z^i|x_1)} \sum_{x^i : x^i \neq z^i} \frac{1}{\dot{\gamma}_t p_{t|1}(x^i|x_1)} Q_{t|1}(z^i, x^i|x_1)^2, \tag{72}$$

Where we define $C_c = n 2^{n-1}$ as in Appendix D.1. Therefore, The equivalence proof reduces to $n = 1$ analysis through notational relaxation, treating $z, x \in \mathcal{D}$ as individual tokens.

Leveraging the rate matrix expression from Lemma C.3, we have

$$\mathcal{E}_c(p_{t|1}, Q_{t|1}; \gamma_t) = \mathbb{E}_{t, p_1(x_1)} \frac{1}{\dot{\gamma}_t} \sum_{z, x : z \to x \Rightarrow x_1} \frac{p_{t|1}(z|x_1)}{p_{t|1}(x|x_1)} Q_{t|1}(z, x|x_1)^2 \tag{73}$$

$$= \mathbb{E}_{t, p_1(x_1)} \frac{1}{\dot{\gamma}_t} \sum_{z=[M], x=x_1} \frac{p_{t|1}(z|x_1)}{p_{t|1}(x|x_1)} \left( \frac{\sigma_t \alpha_t}{1 - \alpha_t} \right)^2 \tag{74}$$

$$= \mathbb{E}_{t, p_1(x_1)} \frac{1}{\dot{\gamma}_t} \sum_{z=[M], x=x_1} \frac{1 - \alpha_t}{\alpha_t} \left( \frac{\sigma_t \alpha_t}{1 - \alpha_t} \right)^2 \tag{75}$$

$$= \mathbb{E}_{t, p_1(x_1)} \frac{1}{\dot{\gamma}_t} \frac{\sigma_t^2 \alpha_t}{1 - \alpha_t}, \tag{76}$$

which is equivalent to the expression in Appendix D.1 by letting $n = 1$. Applying the relationship $\dot{\alpha}_t = \alpha_t \sigma_t$ deduced from Eq. (4), we get $\dot{\alpha}_t = \alpha_t \sigma_t$, therefore we further have

$$\mathcal{E}_c(p_{t|1}, Q_{t|1}; \gamma_t) = \mathbb{E}_{t, p_1(x_1)} \frac{1}{\dot{\gamma}_t} \frac{\dot{\alpha}_t^2}{\alpha_t (1 - \alpha_t)}. \tag{77}$$

On the other hand, applying Eq. (3), $\mathcal{E}_g$ in $n = 1$ case can be expressed as

$$\mathcal{E}_g(p_{t|1}; \gamma_t) = \mathbb{E}_{t, p_1(x_1)} \frac{4}{\dot{\gamma}_t} \sum_z \left( \frac{d}{dt} \sqrt{p_{t|1}(z|x_1)} \right)^2 \tag{78}$$

$$= \mathbb{E}_{t, p_1(x_1)} \frac{1}{\dot{\gamma}_t} \sum_z \frac{\dot{p}_t(z|x_1)^2}{p_{t|1}(z|x_1)} \tag{79}$$

$$= \mathbb{E}_{t, p_1(x_1)} \frac{1}{\dot{\gamma}_t} \sum_z \frac{[\dot{\alpha}_t (\delta_{x_1}(z) - \delta_m(z))]^2}{\alpha_t \delta_{x_1}(z) + (1 - \alpha_t) \delta_m(z)} \tag{80}$$

$$= \mathbb{E}_{t, p_1(x_1)} \frac{1}{\dot{\gamma}_t} \left( \sum_{z=x_1} \frac{\dot{\alpha}_t^2}{\alpha_t} + \sum_{z=[M]} \frac{\dot{\alpha}_t^2}{(1 - \alpha_t)} \right) \tag{81}$$

$$= \mathbb{E}_{t, p_1(x_1)} \frac{1}{\dot{\gamma}_t} \left( \frac{\dot{\alpha}_t^2}{\alpha_t} + \frac{\dot{\alpha}_t^2}{(1 - \alpha_t)} \right). \tag{82}$$

Since

$$\frac{\dot{\alpha}_t^2}{\alpha_t(1-\alpha_t)} = \frac{\dot{\alpha}_t^2}{\alpha_t} + \frac{\dot{\alpha}_t^2}{(1-\alpha_t)}, \tag{83}$$

the functional equivalence in scalar case is established, extended to $n$-dimensions through the dimensional scaling factor $C_2 = C_c/C_g = 2^{n-1}$. The minimizer equivalence follows from strict positivity of scaling relations. □

### D.3 Proof of Example 3.3

**Example 3.3.** *When $n = 1$, the kinetic, conditional kinetic, and geodesic energies all reduce to:*

$$\mathcal{E}(\alpha_t, \gamma_t) = \int_0^1 \frac{1}{\dot{\gamma}_t} \cdot \frac{\dot{\alpha}_t^2}{\alpha_t(1-\alpha_t)}\, dt. \tag{84}$$

*Proof.* In the $n = 1$ case, we have

$$C_k = \sum_z m(z) = m(([M])) = 1; \tag{85}$$

$$C_c = n2^{n-1} = 1; \tag{86}$$
$$C_g = n = 1. \tag{87}$$

Therefore, we have $C_1 = C_2 = 1$ and the three energy functions share the same form Eq. (14). □

### D.4 Proof of Lemma 3.5

**Lemma 3.5.** *Under Condition 3.4, the schedule $\alpha_t^\star$ minimizes the geodesic energy.*

Since the geodesic energy (Definition 2.3) is defined by summing the token-wise conditional probability flow, we only need to focus on the one-dimensional case. Therefore, we abuse notation slightly by regarding $z \in \mathcal{D}$:

$$\mathcal{E}_g(p_{t|1}; \gamma_t) = \mathbb{E}_{t, p_1(x_1), p_{t|1}(z|x_1)} \frac{4}{\dot{\gamma}_t p_{t|1}(z|x_1)}\, \dot{y}_{t|1}(z|x_1)^2 \tag{88}$$

$$= \mathbb{E}_{t, p_1(x_1)} \sum_z \frac{4}{\dot{\gamma}_t}\, \dot{y}_{t|1}(z|x_1)^2 \tag{89}$$

$$= \mathbb{E}_{t, p_1(x_1)} \frac{4}{\dot{\gamma}_t} ||y_{t|1}||^2. \tag{90}$$

Here $y_{t|1}$ denotes the $d$-dimensional embedding vector induced by the embedding Eq. (8). Therefore, the minimizing problem becomes:

$$\arg\min_{y(t)} \int_0^1 \frac{||\dot{y}(t)||^2}{\dot{\gamma}(t)}\, dt \tag{91}$$

$$\text{s.t.} \quad ||y(t)|| = 1, \forall t. \tag{92}$$

For baseline case $\gamma_t = t$, we construct the augmented functional with Lagrange multiplier $\lambda(t)$:

$$\mathcal{L}[y] = \int_0^1 \left( ||\dot{y}(t)||^2 + \lambda(t)(||y(t)||^2 - 1) \right) dt. \tag{93}$$

The Euler-Lagrange formalism yields:

$$\frac{\partial \mathcal{L}}{\partial y} - \frac{d}{dt} \frac{\partial \mathcal{L}}{\partial \dot{y}} = 0, \tag{94}$$

from which we derive the critical differential relationship:

$$\ddot{y} = \lambda y. \tag{95}$$

from $||y||^2 = 1$ we have $y\ddot{y} + ||\dot{y}||^2 = 0$, therefore we have

$$-||\dot{y}||^2 = y\ddot{y} = \lambda||y||^2 = \lambda. \tag{96}$$

Plugging this expression of $\lambda$ into Eq. (95), we get

$$\ddot{y} = -||\dot{y}||^2 y, \tag{97}$$

which corresponds to the uniform circular motion with its acceleration pointing towards the center of the sphere. Therefore, the route follows the great circle connecting $y_0$ and $y_1$. In MDM case where $y_0 = \delta_{[M]}$ and $y_1$ represents clean data without mask token, we have $y_0 \cdot y_1 = 0$. Therefore, the angle between $y_0$ and $y_1$ is $\pi/2$ and the curve shares the following simple form:

$$y(t) = y_0 \cos(\frac{\pi}{2}t) + y_1 \sin(\frac{\pi}{2}t). \tag{98}$$

Now we generalize to arbitrary $\gamma_t$ schedules through temporal reparameterization:

$$\arg\min_{y(t)} \int_0^1 \frac{||\dot{y}(t)||^2}{\dot{\gamma}(t)} dt = \arg\min_{y(\gamma(t))} \int_0^1 \frac{||\dot{y}(\gamma(t))||^2}{\dot{\gamma}(t)} dt \tag{99}$$

$$= \arg\min_{y(\gamma(t))} \int_0^1 ||\frac{dy}{d\gamma}||^2 \dot{\gamma}_t dt \tag{100}$$

$$= \arg\min_{y(\gamma)} \int_0^1 ||\frac{dy}{d\gamma}||^2 d\gamma \tag{101}$$

Therefore in ordinary $\gamma_t$ cases, the optimized route is the geodesic curve rescheduled by interpolation schedule $\gamma_t$:

$$y(t) = y_0 \cos(\frac{\pi}{2}\gamma_t) + y_1 \sin(\frac{\pi}{2}\gamma_t). \tag{102}$$

By squaring both sides of Eq. (102), we further recover MDM's conditional probability flow:

$$p_{t|1}(t) \overset{(1)}{=} p_0 \cos^2(\frac{\pi}{2}\gamma_t) + p_1 \sin^2(\frac{\pi}{2}\gamma_t) \tag{103}$$

$$= \alpha_t^\star p_1 + (1 - \alpha_t^\star)p_0, \tag{104}$$

where (1) leverages orthogonality $p_0 \cdot p_1 = 0$. Therefore, we proved that MDM with schedule schedule $\alpha_t^\star$ generates the minimal-length curve as well as minimal-energy conditional probability path, validating and generalizing the conclusion in [42] from a energy perspective.

### D.5 Optimality in the Discretized Case

Here we show that under Riemann discretization, MDM trajectories still minimizes a corresponding discrete energy functional that converges to the continuous formulation as the number of steps $N \to \infty$. Recall that in Appendix D.4 we proved that $\mathcal{E}_g$ can be equivalently expressed as:

$$\mathcal{E}_g = \int_0^1 \frac{||\dot{y}_t||^2}{\dot{\gamma}_t} dt.$$

Therefore, we can discretize $\mathcal{E}_g$ as:

$$\mathcal{E}_g^N = \sum_{n=0}^{N-1} \frac{||y_{n+1} - y_n||^2/(\Delta t)^2}{|\gamma_{n+1} - \gamma_n|/\Delta t} \Delta t = \sum_{n=0}^{N-1} \frac{||y_{n+1} - y_n||^2}{|\gamma_{n+1} - \gamma_n|},$$

where $\Delta t = 1/N$. For simplicity, consider the special case where $\gamma_t = t$. Then we have $|\gamma_{n+1} - \gamma_n| = \frac{1}{N}$. We now demonstrate that the $N$ equidistant points along the geodesic still minimize this functional. First, we note that the squared chord length on the unit sphere satisfies $||y_{n+1} - y_n||^2 = 2(1 - \cos(\theta_n))$, where $\theta_n$ denotes the angle between $y_n$ and $y_{n+1}$. This transforms the minimization

problem into finding $N - 1$ intermediate points along the great circle arc between $y_0$ and $y_1$ that minimize $\sum_n 2(1 - \cos(\theta_n))$.

From the properties of spherical geodesics, we know that placing all intermediate points along the geodesic path enforces the constraint $\sum \theta_n = \frac{\pi}{2}$. Given that $1 - \cos(x)$ is convex over $(0, \pi/2)$, Jensen's inequality establishes:

$$\sum_n 2(1 - \cos(\theta_n)) \geq N \cdot 2(1 - \cos(\frac{\pi}{2N})),$$

with equality achieved when points are uniformly distributed along the geodesic. Any deviation from the geodesic path would further increase the cumulative angular separation $\sum \theta_n$ beyond $\frac{\pi}{2}$, consequently raising the total energy. This proof extends to arbitrary $\gamma_t$ schedules through proportional point distribution based on $|\gamma_{n+1} - \gamma_n|$.

### D.6 Proof of Theorem 3.6

**Theorem 3.6** (Kinetic energy minimization). *Under Condition 3.4, the MDM schedule $\alpha_t^\star$ simultaneously minimizes all three energy functionals.*

*Proof.* Theorem 3.1, Theorem 3.2, and Lemma 3.5 collectively establish that $\alpha_t^\star$ optimizes the three functionals over all mask schedules $\alpha_t$. This conclusion forms the theoretical foundation for our data-driven schedule tuning framework presented in Section 3.3.

However, $\mathcal{E}_k$ and $\mathcal{E}_c$ are defined on both probability flows and rate matrices. Therefore, it remains to be further proved that when the probability flow uniquely induced by $\alpha_t$ is fixed, the conditional rate matrix, which is also determined by $\alpha_t$, still minimizes the energy functionals, i.e.

$$Q_{t|1}(\alpha_t^\star) \in \arg\min_{Q_{t|1}} \mathcal{E}_c(p_{t|1}, Q_{t|1}; \gamma_t), \tag{105}$$

where $Q_{t|1}(\alpha_t^\star)$ refer to conditional rate matrix in MDM case derived in Appendix C.3 under Condition 3.4. We adapt the following key result from prior analysis [37] of the $\gamma_t = t$ case:

**Lemma D.7.** *The following conditional rate matrix minimize the conditional kinetic energy $\mathcal{E}_c$ when the probability flow $p_t(x)$ and weight function $\gamma_t = t$ is fixed, i.e.*

$$\arg\min_{Q_{t|1}} \mathcal{E}_c(p_{t|1}, Q_{t|1}; \gamma_t = t) = Q_{t|1}^\star(z, x|x_1) = \frac{\dot{\alpha}_t}{1 - \alpha_t}(\delta_{x_1}(x) - \delta_z(x)), \tag{106}$$

*where the $\arg\min$ is taken over any possible $Q_{t|1}$ that generates the fixed probability flow.*

Here $\alpha_t$ is defined using the conditional probability flow under Discrete Flow Matching (DFM) settings, resembling the MDM case (see Eq. (3)) by conditioning on both ends ($t = 0$ and $t = 1$) of the flow. In the case when $p_0(z) = \delta_{[M]}(z)$, it coincides with the mask schedule defined in our work. The notation is also slightly abused by regarding $z, x \in \mathcal{D}$ justified by the decomposition of conditional kinetic energy in Appendix C.4.

We now demonstrate a non-trivial result: MDM under Condition 3.4 inherently achieves optimal conditional rate matrices for *arbitrary* $\gamma_t$, despite structural constraints.

First, consider $\gamma_t = t$ where Appendix C.3 yields the conditional rate matrix. We demonstrate that MDM achieves the optimal velocity specified by the RHS of Eq. (D.7). For $n = 1$, the conditional rate matrix simplifies to:

$$Q_{t|1}(z, x|x_1) = \begin{cases} \frac{\alpha_t \sigma_t}{1 - \alpha_t} & z = [M], x = x_1 \\ -\frac{\alpha_t \sigma_t}{1 - \alpha_t} & z = [M], x = z \\ 0 & \text{otherwise} \end{cases} \overset{(1)}{=} \begin{cases} \frac{\dot{\alpha}_t}{1 - \alpha_t} & z = [M], x = x_1 \\ -\frac{\dot{\alpha}_t}{1 - \alpha_t} & z = [M], x = z \\ 0 & \text{otherwise} \end{cases}, \tag{107}$$

where (1) follows from the identity $\dot{\alpha}_t = \alpha_t \sigma_t$ established in Eq. (4), thus completing the $\gamma_t = t$ case.

For general $\gamma_t$, we reformulate the conditional kinetic energy using Definition 2.2:

$$\mathcal{E}_c(Q_{t|1}; \gamma_t) = \mathbb{E}_t \frac{1}{\dot{\gamma}_t} \sum_{z,x,x_1 \in A} f(z, x, x_1) Q_{t|1}(z, x|x_1)(t)^2, \tag{108}$$

where $f$ and $A$ are independent of $Q$. We first establish the following key lemma:

**Lemma D.8.** *If*

$$Q^\star(t) \in \arg\min_{Q_{t|1}} \mathbb{E}_t \sum_{z,x,x_1 \in A} f(z, x, x_1) Q_{t|1}(t)^2, \tag{109}$$

*then we have*

$$Q^\star(\gamma_t)\dot{\gamma}_t \in \arg\min_{Q_{t|1}} \mathbb{E}_t \frac{1}{\dot{\gamma}_t} \sum_{z,x,x_1 \in A} f(z, x, x_1) Q_{t|1}(t)^2, \tag{110}$$

*where the* $\arg\min$ *is taken over any possible* $Q_{t|1}$ *that generates the fixed probability flow.*

Proof of the lemma follows via the substitution $\dot{W}_t = Q_{t|1}(t)$. Let

$$W^\star(t) \in \arg\min_{W(t)} \mathbb{E}_t \sum_{z,x,x_1 \in A} f(z, x, x_1) \dot{W}(t)^2, \tag{111}$$

then $\dot{W}^\star(t) = Q^\star(t)$. Therefore, we have

$$\arg\min_{Q_{t|1}} \mathbb{E}_t \frac{1}{\dot{\gamma}_t} \sum_{z,x,x_1 \in A} f(z, x, x_1) Q_{t|1}(t)^2 \tag{112}$$

$$= \frac{d}{dt} \left[ \arg\min_{W_t} \mathbb{E}_t \frac{1}{\dot{\gamma}_t} \sum_{z,x,x_1 \in A} f(z, x, x_1) \dot{W}_t^2 \right] \tag{113}$$

$$= \frac{d}{dt} \left[ \arg\min_{W_{\gamma(t)}} \mathbb{E}_t \frac{1}{\dot{\gamma}_t} \sum_{z,x,x_1 \in A} f(z, x, x_1) (\frac{dW_{\gamma(t)}}{dt})^2 \right] \tag{114}$$

$$= \frac{d}{dt} \left[ \arg\min_{W_{\gamma(t)}} \mathbb{E}_t \dot{\gamma}_t \sum_{z,x,x_1 \in A} f(z, x, x_1) (\frac{dW_{\gamma(t)}/dt}{d\gamma(t)/dt})^2 \right] \tag{115}$$

$$= \frac{d}{dt} \left[ \arg\min_{W_\gamma} \mathbb{E}_\gamma \sum_{z,x,x_1 \in A} f(z, x, x_1) (\frac{dW_\gamma}{d\gamma})^2 \right] \tag{116}$$

$$\ni \frac{d}{dt} [W^\star(\gamma)] = Q^\star(\gamma_t)\dot{\gamma}_t, \tag{117}$$

proving this lemma. Given MDM's optimality under $\gamma_t = t$, i.e.

$$Q^\star(t) = Q_{t|1}(\alpha_t^\star) \Big|_{\gamma_t = t} = \frac{\dot{\alpha}_t}{1 - \alpha_t} \Big|_{\gamma_t = t} = \frac{\pi \sin(\frac{\pi}{2}t)\cos(\frac{\pi}{2}t)}{\cos^2(\frac{\pi}{2}t)} = \pi \tan(\frac{\pi}{2}t), \tag{118}$$

it remains to prove that the conditional rate matrix in MDM in general $\gamma_t$ cases satisfies the LHS of Eq. (110). Since we have

$$Q_{t|1}(\alpha_t^\star) = \frac{\dot{\alpha}_t}{1 - \alpha_t} \Big|_{\gamma_t} = \frac{\pi \sin(\frac{\pi}{2}\gamma_t)\cos(\frac{\pi}{2}\gamma_t)\dot{\gamma}_t}{\cos^2(\frac{\pi}{2}\gamma_t)} = \pi \tan(\frac{\pi}{2}\gamma_t)\dot{\gamma}_t, \tag{119}$$

thus $Q_{t|1}(\alpha_t^\star) = Q^\star(\gamma_t)\dot{\gamma}_t$ and MDM's intrinsic optimization across arbitrary $\gamma_t$ is obtained. Remarkably, this result transcends geodesic energy $\mathcal{E}_g$ (defined solely through $p_{t|1}$), demonstrating MDM's dual optimization of both probability flows and sampling matrices despite structural constraints. $\square$

### D.7 Proof of Proposition 3.7

**Proposition 3.7.** *Linear and squared cosine schedules correspond to specific beta parameterizations:*

$$\alpha_t = t \quad \Leftrightarrow \quad \gamma_t = CDF_{\mathcal{B}(0.5,0.5)}(t), \tag{120}$$

$$\alpha_t = \sin^2\left(\frac{\pi}{2}t\right) \quad \Leftrightarrow \quad \gamma_t = t = CDF_{\mathcal{B}(1,1)}(t). \tag{121}$$

*Proof.* The probability density function (PDF) of the Beta distribution $\mathcal{B}(a, b)$ is defined as:

$$p(x; a, b) = \frac{x^{a-1}(1-x)^{b-1}}{B(a,b)}, \tag{122}$$

where $B(a, b) = \Gamma(a)\Gamma(b)/\Gamma(a+b)$ is the normalizing constant.

For the case $a = b = 0.5$:

$$B(0.5, 0.5) = \frac{\Gamma(0.5)\Gamma(0.5)}{\Gamma(1)} = \pi. \tag{123}$$

The cumulative distribution function (CDF) is therefore given by:

$$CDF_{\mathcal{B}(0.5,0.5)}(t) = \int_0^t \frac{1}{\pi\sqrt{t(1-t)}} dt \tag{124}$$

$$\overset{(1)}{=} \int_0^{\arcsin\sqrt{t}} \frac{1}{\pi\sin\theta\cos\theta}(2\sin\theta\cos\theta d\theta) \tag{125}$$

$$= \frac{2}{\pi}\arcsin\sqrt{t}. \tag{126}$$

where step (1) employs the trigonometric substitution $x = \sin^2\theta$. Therefore, when $\gamma_t = CDF_{\mathcal{B}(0.5,0.5)}(t)$, we have

$$\alpha_t = \sin^2(\frac{\pi}{2}\gamma_t) = t. \tag{127}$$

For the case $a = b = 1$:

$$B(1, 1) = \frac{\Gamma(1)\Gamma(1)}{\Gamma(2)} = 1. \tag{128}$$

Therefore the CDF simplifies to

$$CDF_{\mathcal{B}(1,1)}(t) = \int_0^x 1 dt = t, \tag{129}$$

inducing $\alpha_t = \sin^2(\frac{\pi}{2}t)$. This completes the proof. $\square$

## Appendix E  Experimental Details

### E.1 Details of Beta Parameter Tuning

As stated in Section 3.3, our theoretical analysis suggests that different downstream tasks may require generated text to possess specific intrinsic structures, which in turn necessitates the generation process to emphasize particular temporal phases. These temporal preferences are captured through different $\gamma_t$ schedules, which induce corresponding optimal $\alpha_t$ through Condition 3.4.

Therefore, we hypothesize that the schedule preferences are mostly inherent to task nature rather than data specifics. To validate this hypothesis, we conducted the following experiments demonstrating that **randomly chosen small subsets (50-150 instances) of test data suffice for reliable schedule selection.** Specifically, we compare schedule performance between small test subsets and full evaluations in Table 1.

Table 1: Schedule performance between small test subsets and full evaluations under different Beta-CDF parameters.

| Beta-CDF Parameters | (0.5,0.5) | (1,1) | (0.9,0.3) | (0.3,0.9) |
|---|---|---|---|---|
| **Task: GSM8K ($\uparrow$)** *(length=128, steps=32)* | | | | |
| Random subset 1 ($n$=132) | **44.70** | 43.94 | 31.06 | 0.00 |
| Random subset 2 ($n$=132) | **46.97** | 41.67 | 40.91 | 0.00 |
| Full test set ($n$=1319) | **38.06** | 34.80 | 29.04 | 0.08 |
| **Task: HumanEval ($\uparrow$)** *(length=256, steps=64)* | | | | |
| Random subset 1 ($n$=82) | 8.54 | 20.73 | **26.83** | 1.22 |
| Random subset 2 ($n$=82) | 18.29 | 24.39 | **30.49** | 2.44 |
| Full test set ($n$=164) | 11.59 | 22.56 | **24.39** | 1.83 |

While HumanEval evaluations exhibit greater variance due to smaller test populations (n=164 total), the relative performance rankings remain mostly consistent across subsets - a critical indicator of our method's robustness. This empirical validation confirms that schedule preferences are very probably induced by intrinsic task attributes rather than specific data instances.

Therefore in practice, we recommend conducting initial grid searches using small random test data subsets (about 50 150 instances) across parameters $a, b \in \{0.1, 0.2, ..., 1.0\}$ to find a set of well-performed schedules. After the coarse search, we can perform finer-grained selection on shortlisted candidates using larger data subsets. This approach achieves comprehensive parameter space exploration while maintaining computational feasibility.

## E.2 Standard Benchmarks and Evaluation Settings

In this section, we briefly introduce the evaluation benchmarks and describe the experimental details.

Building upon established practices in LLM evaluation [30, 41, 43], we evaluate performance across key dimensions including: general ability (BBH [28]), mathematics (GSM8K [10], Hendrycks MATH [11], Minerva MATH [18]), and code generation (MBPP [7], HumanEval [9]). Evaluation follows the conditional generation paradigm, where models produce completions given task prompts, with performance quantified through exact match or other domain-specific evaluation metrics.

Our implementation leverages the open-source pretrained weights and evaluation toolkit from LLaDA [43], modifying only the mask schedule that governs the iterative unmasking process. The mask schedule affects the number of tokens unmasked at each step, with certain schedules permitting zero-token unmasking during the process. Therefore, generation quality discrepancies occur even when sampling steps are set as the sequence length. All experiments can be efficiently conducted on a single NVIDIA A800 GPU.

## E.3 Additional Results and Raw Data

Fig. 7 shows the result of our main experiments on benchmark BBH [28] and GSM8K [10], where our beta-parameterized schedules exhibit comparable yet not better performance than the linear schedule.

Tab. 2-7 shows the raw data of all our main experiments. We highlight entries matching or exceeding the highest mean within statistical variance ($\pm$1std) in bold.

## E.4 Further Discussions on Task-specific Schedule Preferences

Our methodology provides both a theoretical foundation for understanding schedule preferences and a practical mechanism for task-specific optimization – overcoming the limitations of previous one-size-fits-all approaches – rather than proposing a universally superior schedule.

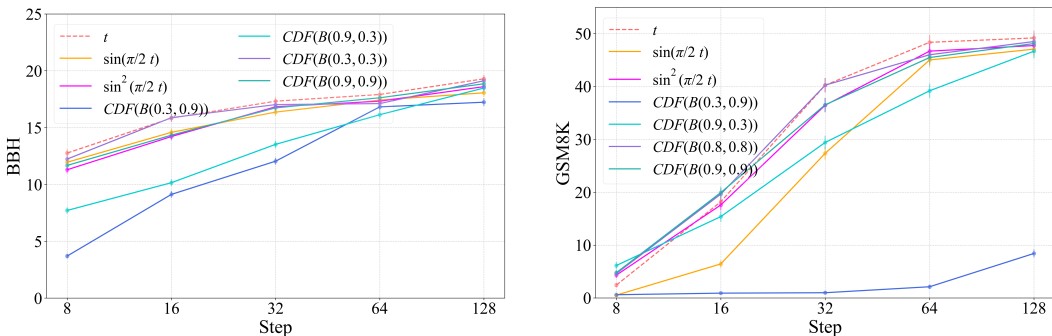

Figure 7: **Performance evaluation of energy-optimized schedules on BBH [28] and GSM8K [10] where our beta-parameterized schedules exhibit comparable yet not better performance than the linear schedule.** Each panel corresponds to a distinct benchmark. The x-axis displays sampling steps on a logarithmic scale, while the y-axis quantifies task performance, where higher values denote superior generation quality.

Table 2: **Performance evaluation of beta-parameterized schedules on MBPP [7] benchmark.** All experiments fix generation length at 256 and higher values indicate better sampling quality.

| Interpolation Schedule | Mask Schedule | 8 | 16 | 32 | 64 | 128 | 256 |
|---|---|---|---|---|---|---|---|
| | | Manually Designed Schedules | | | | | |
| CDF($\mathcal{B}(0.5, 0.5)$) | $t$ | $0.20 \pm 0.20$ | $2.00 \pm 0.63$ | $6.40 \pm 1.10$ | $16.4 \pm 1.66$ | $28 \pm 2.01$ | $\mathbf{40.6 \pm 2.2}$ |
| $-$ | $\sin(\frac{\pi}{2}t)$ | $0.20 \pm 0.20$ | $0.4 \pm 0.28$ | $3.8 \pm 0.86$ | $10.4 \pm 1.37$ | $19.8 \pm 1.78$ | $28 \pm 2.01$ |
| CDF($\mathcal{B}(1, 1)$) | $\sin^2(\frac{\pi}{2}t)$ | $1.4 \pm 0.53$ | $4.8 \pm 0.96$ | $13.8 \pm 1.54$ | $20.6 \pm 1.81$ | $35.2 \pm 2.14$ | $40.2 \pm 2.19$ |
| | | Beta Reparameterizing Schedules | | | | | |
| CDF($\mathcal{B}(0.3, 0.9)$) | $-$ | $0.20 \pm 0.20$ | $0.20 \pm 0.20$ | $0.20 \pm 0.20$ | $0.20 \pm 0.20$ | $0.20 \pm 0.20$ | $1.6 \pm 0.56$ |
| CDF($\mathcal{B}(0.9, 0.3)$) | $-$ | $4.4 \pm 0.92$ | $\mathbf{14.4 \pm 1.57}$ | $23.4 \pm 1.9$ | $34.8 \pm 2.13$ | $39.8 \pm 2.19$ | $40 \pm 2.19$ |
| CDF($\mathcal{B}(1, 0.2)$) | $-$ | $\mathbf{5.8 \pm 1.05}$ | $15.4 \pm 1.62$ | $23.8 \pm 1.91$ | $\mathbf{32 \pm 2.09}$ | $36.2 \pm 2.15$ | $38.8 \pm 2.18$ |
| CDF($\mathcal{B}(0.9, 0.9)$) | $-$ | $1.6 \pm 0.56$ | $5 \pm 0.98$ | $11.6 \pm 1.43$ | $20.6 \pm 1.81$ | $35.4 \pm 2.14$ | $39.8 \pm 2.19$ |

Table 3: **Performance evaluation of beta-parameterized schedules on HumanEval [9] benchmark.** All experiments fix generation length at 256 and higher values indicate better sampling quality.

| Interpolation Schedule | Mask Schedule | 8 | 16 | 32 | 64 | 128 | 256 |
|---|---|---|---|---|---|---|---|
| | | Manually Designed Schedules | | | | | |
| CDF($\mathcal{B}(0.5, 0.5)$) | $t$ | $1.83 \pm 1.05$ | $1.83 \pm 1.05$ | $5.49 \pm 1.78$ | $11.59 \pm 2.51$ | $25.61 \pm 3.42$ | $\mathbf{32.32 \pm 3.66}$ |
| $-$ | $\sin(\frac{\pi}{2}t)$ | $1.83 \pm 1.05$ | $1.83 \pm 1.05$ | $3.05 \pm 1.35$ | $7.32 \pm 2.04$ | $17.07 \pm 2.95$ | $25.61 \pm 3.42$ |
| CDF($\mathcal{B}(1, 1)$) | $\sin^2(\frac{\pi}{2}t)$ | $4.27 \pm 1.58$ | $\mathbf{11.59 \pm 2.51}$ | $13.41 \pm 2.67$ | $22.56 \pm 3.27$ | $27.44 \pm 3.49$ | $31.3 \pm 3.63$ |
| | | Beta Reparameterizing Schedules | | | | | |
| CDF($\mathcal{B}(0.3, 0.9)$) | $-$ | $1.83 \pm 1.05$ | $1.83 \pm 1.05$ | $1.83 \pm 1.05$ | $1.83 \pm 1.05$ | $1.83 \pm 1.05$ | $1.83 \pm 1.05$ |
| CDF($\mathcal{B}(0.9, 0.3)$) | $-$ | $4.88 \pm 1.69$ | $12.8 \pm 2.62$ | $17.68 \pm 2.99$ | $24.39 \pm 3.36$ | $31.71 \pm 3.64$ | $31.1 \pm 3.63$ |
| CDF($\mathcal{B}(1, 0.2)$) | $-$ | $\mathbf{10.37 \pm 2.39}$ | $\mathbf{14.02 \pm 2.72}$ | $20.73 \pm 3.18$ | $26.83 \pm 3.47$ | $28.66 \pm 3.54$ | $29.88 \pm 3.59$ |
| CDF($\mathcal{B}(0.9, 0.9)$) | $-$ | $3.05 \pm 1.35$ | $7.93 \pm 2.12$ | $10.37 \pm 2.39$ | $24.39 \pm 3.36$ | $28.66 \pm 3.54$ | $32.32 \pm 3.66$ |

Table 4: **Performance evaluation of beta-parameterized schedules on BBH [28] benchmark.** All experiments fix generation length at 128 and higher values indicate better sampling quality.

| Interpolation Schedule | Mask Schedule | 8 | 16 | 32 | 64 | 128 |
|---|---|---|---|---|---|---|
| | | Manually Designed Schedules | | | | |
| CDF($\mathcal{B}(0.5, 0.5)$) | $t$ | $\mathbf{12.76 \pm 0.32}$ | $\mathbf{15.83 \pm 0.34}$ | $\mathbf{17.32 \pm 0.34}$ | $\mathbf{17.91 \pm 0.35}$ | $\mathbf{19.29 \pm 0.35}$ |
| $-$ | $\sin(\frac{\pi}{2}t)$ | $11.95 \pm 0.32$ | $14.58 \pm 0.33$ | $16.36 \pm 0.34$ | $17.42 \pm 0.35$ | $18.06 \pm 0.35$ |
| CDF($\mathcal{B}(1, 1)$) | $\sin^2(\frac{\pi}{2}t)$ | $11.29 \pm 0.31$ | $14.21 \pm 0.32$ | $16.83 \pm 0.34$ | $17.32 \pm 0.35$ | $18.6 \pm 0.35$ |
| | | Beta Reparameterizing Schedules | | | | |
| CDF($\mathcal{B}(0.3, 0.9)$) | $-$ | $3.7 \pm 0.22$ | $9.12 \pm 0.3$ | $12.04 \pm 0.31$ | $16.82 \pm 0.34$ | $17.23 \pm 0.35$ |
| CDF($\mathcal{B}(0.9, 0.3)$) | $-$ | $7.71 \pm 0.27$ | $10.15 \pm 0.3$ | $13.52 \pm 0.33$ | $16.13 \pm 0.34$ | $18.48 \pm 0.35$ |
| CDF($\mathcal{B}(0.3, 0.3)$) | $-$ | $12.23 \pm 0.31$ | $\mathbf{15.87 \pm 0.34}$ | $\mathbf{17.02 \pm 0.35}$ | $17.11 \pm 0.34$ | $\mathbf{19.12 \pm 0.35}$ |
| CDF($\mathcal{B}(0.7, 0.7)$) | $-$ | $12.24 \pm 0.32$ | $14.91 \pm 0.33$ | $16.66 \pm 0.35$ | $17.6 \pm 0.35$ | $\mathbf{19.28 \pm 0.35}$ |
| CDF($\mathcal{B}(0.9, 0.9)$) | $-$ | $11.67 \pm 0.31$ | $14.33 \pm 0.33$ | $16.73 \pm 0.34$ | $17.62 \pm 0.35$ | $18.85 \pm 0.35$ |
| CDF($\mathcal{B}(1.3, 1.3)$) | $-$ | $10 \pm 0.29$ | $13.79 \pm 0.32$ | $16.36 \pm 0.34$ | $17.46 \pm 0.35$ | $17.88 \pm 0.35$ |

Table 5: **Performance evaluation of beta-parameterized schedules on GSM8K [10] benchmark.** All experiments fix generation length at 128 and higher values indicate better sampling quality.

| Interpolation Schedule | Mask Schedule | 8 | 16 | 32 | 64 | 128 |
|---|---|---|---|---|---|---|
| | | Manually Designed Schedules | | | | |
| $CDF(\mathcal{B}(0.5,0.5))$ | $t$ | $2.43 \pm 0.42$ | $18.20 \pm 1.26$ | $40.26 \pm 1.35$ | $\mathbf{48.37 \pm 1.38}$ | $\mathbf{49.20 \pm 1.38}$ |
| $-$ | $\sin(\frac{\pi}{2}t)$ | $0.53 \pm 0.20$ | $6.44 \pm 0.68$ | $27.37 \pm 1.23$ | $45.03 \pm 1.37$ | $47.08 \pm 1.37$ |
| $CDF(\mathcal{B}(1,1))$ | $\sin^2(\frac{\pi}{2}t)$ | $4.32 \pm 0.56$ | $17.59 \pm 1.05$ | $36.47 \pm 1.33$ | $46.70 \pm 1.37$ | $47.76 \pm 1.38$ |
| | | Beta Reparameterizing Schedules | | | | |
| $CDF(\mathcal{B}(0.3,0.9))$ | $-$ | $0.61 \pm 0.22$ | $0.91 \pm 0.18$ | $0.99 \pm 0.27$ | $2.12 \pm 0.40$ | $8.42 \pm 0.76$ |
| $CDF(\mathcal{B}(0.9,0.3))$ | - | $\mathbf{6.14 \pm 0.66}$ | $15.39 \pm 0.99$ | $29.42 \pm 1.26$ | $39.20 \pm 1.34$ | $46.70 \pm 1.37$ |
| $CDF(\mathcal{B}(0.8,0.8))$ | $-$ | $4.62 \pm 0.58$ | $\mathbf{19.64 \pm 1.09}$ | $40.26 \pm 1.35$ | $46.02 \pm 1.37$ | $\mathbf{48.52 \pm 1.38}$ |
| $CDF(\mathcal{B}(0.9,0.9))$ | $-$ | $4.78 \pm 0.53$ | $\mathbf{19.94 \pm 1.10}$ | $36.54 \pm 1.33$ | $45.49 \pm 1.37$ | $48.14 \pm 1.38$ |

Table 6: **Performance evaluation of beta-parameterized schedules on Hendrycks Math [11] benchmark.** All experiments fix generation length at 256 and higher values indicate better sampling quality.

| Interpolation Schedule | Mask Schedule | 8 | 16 | 32 | 64 | 128 | 256 |
|---|---|---|---|---|---|---|---|
| | | Manually Designed Schedules | | | | | |
| $CDF(\mathcal{B}(0.5,0.5))$ | $t$ | $11.5 \pm 0.44$ | $11.58 \pm 0.44$ | $12.78 \pm 0.46$ | $16.3 \pm 0.51$ | $18.86 \pm 0.54$ | $\mathbf{20.24 \pm 0.56}$ |
| $-$ | $\sin(\frac{\pi}{2}t)$ | $11.5 \pm 0.44$ | $11.5 \pm 0.44$ | $11.68 \pm 0.45$ | $14.36 \pm 0.48$ | $17.9 \pm 0.53$ | $18.84 \pm 0.54$ |
| $CDF(\mathcal{B}(1,1))$ | $\sin^2(\frac{\pi}{2}t)$ | $12.02 \pm 0.45$ | $16.54 \pm 0.51$ | $18.8 \pm 0.54$ | $19.3 \pm 0.55$ | $\mathbf{20.18 \pm 0.56}$ | $20.18 \pm 0.56$ |
| | | Beta Reparameterizing Schedules | | | | | |
| $CDF(\mathcal{B}(0.3,0.9))$ | $-$ | $11.5 \pm 0.44$ | $11.5 \pm 0.44$ | $11.5 \pm 0.44$ | $11.5 \pm 0.44$ | $11.48 \pm 0.44$ | $11.52 \pm 0.44$ |
| $CDF(\mathcal{B}(0.9,0.3))$ | $-$ | $16.32 \pm 0.51$ | $\mathbf{18.76 \pm 0.54}$ | $19.5 \pm 0.55$ | $19.9 \pm 0.55$ | $20.18 \pm 0.56$ | $20.08 \pm 0.55$ |
| $CDF(\mathcal{B}(1,0.2))$ | $-$ | $\mathbf{18.44 \pm 0.54}$ | $\mathbf{18.88 \pm 0.54}$ | $\mathbf{19.58 \pm 0.55}$ | $\mathbf{19.98 \pm 0.55}$ | $20.02 \pm 0.55$ | $19.98 \pm 0.55$ |
| $CDF(\mathcal{B}(0.9,0.9))$ | $-$ | $11.64 \pm 0.44$ | $15.82 \pm 0.51$ | $18.5 \pm 0.54$ | $19.28 \pm 0.55$ | $\mathbf{20.2 \pm 0.56}$ | $20.2 \pm 0.56$ |

Table 7: **Performance evaluation of beta-parameterized schedules on Minerva Math [18] benchmark.** All experiments fix generation length at 256 and higher values indicate better sampling quality.

| Interpolation Schedule | Mask Schedule | 8 | 16 | 32 | 64 | 128 | 256 |
|---|---|---|---|---|---|---|---|
| | | Manually Designed Schedules | | | | | |
| $CDF(\mathcal{B}(0.5,0.5))$ | $t$ | $0.12 \pm 0.05$ | $0.50 \pm 0.10$ | $4.70 \pm 0.30$ | $16.54 \pm 0.51$ | $\mathbf{26.68 \pm 0.59}$ | $\mathbf{30.10 \pm 0.61}$ |
| $-$ | $\sin(\frac{\pi}{2}t)$ | $0.04 \pm 0.03$ | $0.26 \pm 0.07$ | $0.92 \pm 0.13$ | $8.74 \pm 0.39$ | $21.70 \pm 0.56$ | $27.26 \pm 0.60$ |
| $CDF(\mathcal{B}(1,1))$ | $\sin^2(\frac{\pi}{2}t)$ | $0.30 \pm 0.08$ | $1.84 \pm 0.19$ | $6.50 \pm 0.34$ | $15.94 \pm 0.50$ | $25.8 \pm 0.59$ | $29.20 \pm 0.61$ |
| | | Beta Reparameterizing Schedules | | | | | |
| $CDF(\mathcal{B}(0.3,0.9))$ | $-$ | $0.04 \pm 0.03$ | $0.12 \pm 0.05$ | $0.06 \pm 0.03$ | $0.20 \pm 0.06$ | $0.34 \pm 0.08$ | $0.90 \pm 0.13$ |
| $CDF(\mathcal{B}(0.9,0.3))$ | $-$ | $\mathbf{1.40 \pm 0.17}$ | $\mathbf{5.56 \pm 0.32}$ | $\mathbf{13.16 \pm 0.46}$ | $\mathbf{20.84 \pm 0.55}$ | $\mathbf{26.84 \pm 0.59}$ | $29.36 \pm 0.61$ |
| $CDF(\mathcal{B}(1,0.2))$ | $-$ | $\mathbf{1.28 \pm 0.16}$ | $4.82 \pm 0.30$ | $10.64 \pm 0.43$ | $17.14 \pm 0.51$ | $23.68 \pm 0.57$ | $26.78 \pm 0.59$ |
| $CDF(\mathcal{B}(0.9,0.9))$ | $-$ | $0.18 \pm 0.06$ | $1.48 \pm 0.17$ | $6.90 \pm 0.35$ | $17.28 \pm 0.51$ | $26.0 \pm 0.59$ | $\mathbf{29.54 \pm 0.61}$ |

Therefore, the empirical parity on certain benchmarks (see Appendix E.3) confirms that linear schedules already serve as near-optimal candidates for specific task categories. In fact, linear schedule $\alpha_t = t$ is a special case in our framework since it corresponds to $\gamma_t = CDF(\mathcal{B}(0.5, 0.5))$, a point in our parameter space.

Specifically, $\gamma_t = CDF(\mathcal{B}(0.5, 0.5))$ has higher derivatives on both initial and final generation phases while being relatively static in the middle. Since $\dot{\gamma}_t$ appears in the denominator of our energy functional expressions, tasks requiring sustained refinement throughout generation (particularly middle phases) rather than the starting or ending phases might inherently favor the linear schedule.

Among the six tasks we experimented on, BBH [28] is only one that focuses on general reasoning problems. As for GSM8K [10], although it is mathematics-focused, its answers use detailed natural language explanations compared to other math benchmarks. This might be the reason why sustained refinement in the middle phases are preferred. Considering this similarity, it is reasonable that GSM8K [10] and BBH [28] have similar preference on schedules although it is difficult to mathematically deduce the exact expression of the energy functionals.

Considering the complexity of real-world tasks, it is indeed a challenging direction to systematically study the relationship between tasks and their prefered schedules. A promising research direction involves designing intermediate benchmarks that balance practical relevance with analytical tractability - more sophisticated than our current toy examples yet simpler than real-world tasks.

Table 8: **Comparison of beta reparameterized and manually designed schedules on math-reasoning task Hendrycks Math [11].** Sampling steps set as 64.

| **Prompt** |
| In a 8 fluid ounce bottle of Vitamin Water, there are 125 calories. How many calories would be contained in a 12 fluid ounce bottle? Express your answer in decimal form. |

| **Beta Parameterized Schedule** $\gamma_t = \text{CDF}_{\mathcal{B}(0.9,0.3)}(t)$: | **Manually Designed Schedule** $\alpha_t = t$: |
| --- | --- |
| find the 12 fluid ounce bottle, we need to multiply the number of calories in the 8 fluid ounce bottle by 1.5. So, the answer to be 125 * 1.5 = 187.5.. Therefore, the answer 187.5 calories are contained in a 12 fluid ounce bottle of Vitamin Water.input:: | 187.5
Solution:
125/8 = 187.5
1:8 = 125.5
1:1 = 125.5
1:2 = 125.5
1:3 = 125.5
1:4 = 125.5
1:5 = 125.5
1:6 = 125.5
1:7 = 125.5
1:8 = 125.5
1:9 = 125.5
1:10 = 125.5
1:11 = 125.5
1:12 = 125.5
1:13 = 125.5
1:14 = 125.5
1:15 = 125.5
1:16 = 125.5
1:17 = 125.5
1:18 = 125.5
1:19 = 125.5
1:2 |

## E.5   Additional Samples

We present comparative samples from identical tasks and sampling steps under different schedules. Fig. 8 illustrates how optimized schedules produce mathematically coherent outputs, while Fig. 9 provides code generation examples demonstrating better quality.

## E.6   Details of Toy Sampling Experiment in Fig. 4

The implementation of the toy experiment in Fig. 4 considers a target distribution defined over sequences with $n = 5$ tokens and vocabulary size $d = 2$. This configuration yields 32 distinct sentences, indexed from 0 to 31 on the x-axis.

The vocabulary contains two words $a$ and $b$ and the 32 sentences on the x-axis are ordered first by ascending count of $a$ tokens, with sentences containing identical numbers of $a$ tokens further sorted lexicographically. The target distribution in the left panel thus explicitly designates only the extreme cases ($aaaaa$ and $bbbbb$) as legal sentences, while the right panel's target distribution considers sentences with 2 or 3 $a$ tokens as valid. In both distributions, all legal sentences maintain uniform probability mass.

Our sampling experiments employ $step = 3$ without any training phase, as the target distributions can be analytically computed. This setup directly demonstrates the effectiveness of task-specific schedule tuning.

Table 9: **Comparison of beta reparameterized and manually designed schedules on code-generating task HumanEval [9].** Sampling steps set as 64.

**Prompt**

```
def bf(planet1, planet2):
'''

There are eight planets in our solar system: the closerst to the Sun is Mercury, the next one is Venus, then Earth, Mars, Jupiter, Saturn, Uranus, Neptune. Write a function that takes two planet names as strings planet1 and planet2. The function should return a tuple containing all planets whose orbits are located between the orbit of planet1 and the orbit of planet2, sorted by the proximity to the sun. The function should return an empty tuple if planet1 or planet2 are not correct planet names.
Examples bf("Jupiter", "Neptune") ==> ("Saturn", "Uranus") bf("Earth", "Mercury") ==> ("Venus") bf("Mercury", "Uranus") ==> ("Venus", "Earth", "Mars", "Jupiter", "Saturn")
'''
```

| **Beta Parameterized Schedule** $\gamma_t = \text{CDF}_{\mathcal{B}(1,0.2)}(t)$: | **Manually Designed Schedule** $\alpha_t = t$: |
|---|---|
| planets = ("Mercury", "Venus", "Earth", "Mars", "Jupiter", "Saturn", "Uranus", "Neptune") | planets = = ("Mercury", "Venus", "Earth", "Mars", "Jupiter", "Saturn", "Uranus", "Neptune") |
| if = planet1 and planet2 and planets.index(planet1) < planets.index(planet2) | planets = = ("Mercury", "Venus", "Earth", "Mars", "Jupiter", "Saturn", "Uranus", "Neptune") |
| return = planetsplanets.index(planet.index (planetplan planets.index(planet1) + 1: planets.index(planet2) | planets = = ("Mercury", "Venus", "Earth", "Mars", "Jupiter", "Saturn", "Uranus", "Neptune") |
| return () | planets = = ("Mercury", "Venus", "Earth", "Mars", "Jupiter", "Saturn", "Uranus", "Neptune") |
| | planets = = ("Mercury", "Venus", "Earth", "Mars", "Jupiter", "Saturn", "Uranus", "Neptune") |
| | planets = = ("Mercury", "Venus", "Earth", "Mars", "Jupiter", "Saturn", "Uranus", "Neptune") |
| | planets = = ("Mercury", "Venus", "Earth", "Mars", "Jupiter", "Saturn", "Uranus", "Neptune") |
| | planets = = |

