# OpenReview forum: "Masked Diffusion Models as Energy Minimization"
_NeurIPS.cc/2025/Conference — NeurIPS 2025 poster_

### Official Review · Reviewer_gvvr · 2025-06-24

**Clarity:** 3
**Significance:** 2
**Originality:** 2
**Rating:** 4
**Confidence:** 4

**Summary:**

The authors propose to examine the kinetic energy of Masked Diffusion Models and to explore the influence of the masking process schedule on it. Three formulations of kinetic energy were proposed and it was proven that they are equivalent up to a multiplicative constant and can be simultaneously minimized w.r.t. masking schedule. Given this insight, an explicit masking schedule solution for the mask kinetic energy minimization was derived. However, this explicit masking schedule solution depends on the time-dependent energy weights $\gamma_t$ that define the kinetic energy functional.

On the practical side of the paper, the authors suggest to parameterize the time-dependent energy weights  $\gamma_t$ as a CDF of the Beta distribution then optimize over the Beta distribution parameters and receive the masking schedule as an explicit kinetic energy minimization solution given the optimized time-dependent energy weights  $\gamma_t$. The practical algorithm is tested on LLaDA 8B [1] masked diffusion model in a post-training manner on a set of general reasoning, mathematical problem solving, and code generation benchmarks.

**Questions:**

- Although I think the presented benchmarks are relatively solid it would be nice to the see evaluation of the proposed method for the optimization of masking schedules of basic MDLM [2] on unconditional generation problems and measure test ppl (e.g. Figure 1 [2]).
- It would be nice to see the full end-to-end algorithm and corresponding evaluations for tuning the masking schedule using the proposed parameterization.
- In both subfigures at Figure 2 there are quite sharp bound after which energy start to drastically grow (green to purple edge). Why is that?
- In Figure 2 caption the following said “Axes represent the beta-parameterization of $\alpha_t$ (see Sec. 3.3)”. What exactly does it mean, what is beta-parameterization of $\alpha_t$ ? I could not find such information in Sec. 3.3
- In Def 2.1 the *unmasking* process rate matrix is used, while as I understand it there also could be *masking* process rate matrix, am I right? If that is the case and in general these energy formulations (Def 2.1, Def 2.2, Def 2.3) can be defined in both ways (forward and backward) I think it should be mentioned somewhere in the main part of the paper.

**Ethical Concerns:**

["NO or VERY MINOR ethics concerns only"]

**Final Justification:**

The authors have addressed most of my concerns.  However, there is still a gap between theory and practice, since authors have not explored counterparametrizations, not involving $\gamma_t$, and hence it is not particularly clear wherever the energy parameterization is necessary for good practical performance. Overall, I find the theory a bit unconvincing, but the practical benefits are valuable regardless.

**Limitations:**

Yes

**Paper Formatting Concerns:**

No  major formatting issues in this paper

**Quality:**

3

**Strengths And Weaknesses:**

Strengths:

- Novel theoretical insights. Interesting and novel theory investigations of the kinetic energy in the light of masking schedule. Closed-form masking schedule minimizer for the kinetic energy gives valuable insight into the design of masked diffusion.
- Simple yet effective method. The suggested method for the post-training schedule change is rather simple and, according to experimental results, effective.
- Relatively robust evaluation. The experimental evaluation is done on the state-of-the-art model LLaDA 8B [1] and six diverse benchmarks, which seem to be a relatively robust evaluation.

Weaknesses:

- Unnecessarily detailed theory. A lot of space in the paper is dedicated to kinetic energy and related theory but in the end, the conclusion is rather simple. Not sure that all that theory is particularly needed to be presented in the main part of the paper given that it doesn’t provide any other insight besides optimal masking schedule form.
- Shifted complexity. The optimal schedule still hinges on time-dependent energy weights $\gamma_t$, effectively transferring the optimization challenge from  $\alpha_t$ to $\gamma_t$ without simplifying the overall problem.
- Missing comparison of parameterizations. In the light of previous weakness, it would be nice to see the comparison of the $\alpha_t$ parametrization as an explicit minimizer of the kinetic energy given energy weights $\gamma_t$, which in turn are parametrized as CDF of Beta distribution and parametrization of $\alpha_t$ directly as CDF of Beta distribution.
- Limited impact scope. With contributions split between a bit of theoretical material and a rather simple post-training optimization of masking schedule, I think the overall paper impact may be limited.
- Lack of a practical algorithm. Seems like the paper explores how various schedules influence MDM inference but stops short of delivering a concrete end-to-end procedure for optimizing the proposed masking schedule $\alpha_t$ parameterization. Would be nice to see one.

[1] Nie, S., Zhu, F., You, Z., Zhang, X., Ou, J., Hu, J., ... & Li, C. (2025). Large language diffusion models. *arXiv preprint arXiv:2502.09992*.

[2] Sahoo, S., Arriola, M., Schiff, Y., Gokaslan, A., Marroquin, E., Chiu, J., ... & Kuleshov, V. (2024). Simple and effective masked diffusion language models. *Advances in Neural Information Processing Systems*, *37*, 130136-130184.

---

> ### Author Rebuttal · Authors · 2025-07-31
>
> Thank you for your extremely thorough review and constructive suggestions. Below is our point-by-point response.
>
> ### Weakness 1: Unnecessarily detailed theory
>
> We appreciate this critical perspective. We would like to clarify that although our optimal schedule induced by Condition 3.4 is rather simple, our theoretical framework serves crucial purposes beyond deriving the optimal schedule.
>
> Specifically, MDM's constrained architecture - where both $p_t$ and $Q_t$ are simultaneously determined by $\alpha_t$ - naturally raises questions about its capacity to achieve optimal transport compared to DFM's decoupled optimization. Through Theorem 3.6, we demonstrate that MDM intrinsically encodes optimal rate matrices within its structural limitations. This theoretical guarantee resolves fundamental doubts about MDM's capabilities while explaining its empirical success, thereby strengthening confidence in the framework's mathematical soundness.
>
> Therefore, we believe that our theoretical analysis establishes foundational insights into MDM's operational mechanisms, being helpful in guiding future works.
>
> ### Weakness 2 & 3: Shifted complexity & Missing comparison of parameterizations
>
> Thank you again for the critical perspective. We would like to clarify the importance of introducing $\gamma_t$ empirically.
>
> First, The $\alpha_t$-$\gamma_t$ duality provides **complementary perspectives** for schedule design. Common $\alpha_t$s correspond to complex $\gamma_t$s without closed-form expressions (e.g. $\alpha_t=\sin(\frac{\pi}{2}t)$ corresponds to $\gamma_t$ that do not have a closed form). Conversely, Beta-CDF parameterized $\gamma_t$s generally yield non-trivial $\alpha_t$s (e.g. $\gamma_t=CDF(\mathcal{B}(0.9,0.3))$ correspond to nontrivial $\alpha_t$).
>
> More importantly, **our empirically effective Beta-CDF tuning method stems directly from the energy perspective**. The discovery that standard schedules (linear/sine-squared) correspond to CDF of Beta distribution (Proposition 3.7) motivated systematic Beta-CDF exploration. This connection remains obscure when operating purely in $\alpha_t$ space.
>
> We appreciate your idea in directly parametrizing $\alpha_t$, and although such parameterization might be able to yield empirical improvements as well, the introduction of $\gamma_t$ is still indispensible due to the reasons above.
>
> We hope that our clarification above has addressed your problem.
>
> ### Weakness 4: Limited impact scope
>
> As clarified in our response to weakness 1-3, despite our rather simple closed-form expression, our theoretical analysis induces unique and nontrivial insights. Also, our effective schedule tuning method, especially the Beta-CDF parameterizing method cannot be done without our theoretical framework. Moreover, as detailed in the response to your next question, we have proposed a more thorough explanation on how to effectively tune the parameters in real-world applications. Such method are also highly motivated by our energy perspective.
>
> Considering these clarifications, we kindly ask you to reassess the significance of our work.
>
> ### Weakness 5 & Question 2: Lack of a practical algorithm
>
> We appreciate this insightful question. Our theoretical analysis suggests that different downstream tasks may require generated text to possess specific intrinsic structures, which in turn necessitates the generation process to emphasize particular temporal phases. These temporal preferences are captured through different $\gamma_t$ schedules, which induce corresponding optimal $\alpha_t$ through Condition 3.4.
>
> Therefore, we hypothesize that the schedule preferences are mostly inherent to task nature rather than data specifics. To validate this hypothesis, we conducted the following experiments demonstrating that **randomly chosen small subsets (50-150 instances) of test data suffice for reliable schedule selection.** Specifically, we compare schedule performance between small test subsets and full evaluations (generation length=256, steps=64):
> | Beta-CDF Parameters | (0.5,0.5) | (1,1) | (0.9,0.3) | (0.3,0.9) |
> | :--- | :---: | :---: |:---: | :---: |
> | **task=GSM8K($\uparrow$)** |||||
> | Random subset 1 (n=132)| **56.06**  | 53.79 | 52.27 | 0.00 |
> | Random subset 2 (n=132)| **55.30**  | 52.27 | 50.76 | 0.76 |
> | Full test set (n=1319) | **52.69** | 50.72 | 51.86 | 0.45 |
> | **task=humaneval($\uparrow$)** |||||
> | Random subset 1 (n=82) | 17.07 | 35.37 | **43.90** | 3.66 |
> | Random subset 2 (n=82)| 6.10 | 15.85 | **19.51** | 2.44 |
> | Full test set (n=164) | 11.59 | 22.56 | **24.39** | 1.83 |
>
> While HumanEval evaluations exhibit greater variance due to smaller test populations (n=164 total), the relative performance rankings remain consistent across subsets - a critical indicator of our method's robustness. This empirical validation confirms that schedule preferences are very probably induced by intrinsic task attributes rather than specific data instances.
>
> Therefore in practice, we recommend conducting initial grid searches using small random test data subsets (about 50~150 instances) across parameters $a,b \in \\{0.1, 0.2, ..., 1.0\\}$ to find a set of well-performed schedules. After the coarse search, we can perform finer-grained selection on shortlisted candidates using larger data subsets. This approach achieves comprehensive parameter space exploration while maintaining computational feasibility.
>
> We will expand this discussion in Section 4 to better guide practitioners in adapting our methodology. We appreciate this valuable question that helps clarify implementation details.
>
> ### Question 1:
>
> > Although I think the presented benchmarks are relatively solid it would be nice to the see evaluation of the proposed method for the optimization of masking schedules of basic MDLM [2] on unconditional generation problems and measure test ppl (e.g. Figure 1 [2]).
>
> Thank you for this question. Our extended experiments demonstrate the broader applicability of our schedule optimization framework to unconditional generation tasks. We still conducted evaluations using Llada due to its unprecedented 8B scale with generation length fixed at 256 tokens and sampling steps at 64. Perplexity measurements were computed via Quen2-7B ([1*]) on 100 generated samples per schedule configuration:
>
> | Beta-CDF Parameters | (0.5,0.5) | (1,1) | (0.9,0.3) | (0.3,0.9) |
> | :--- | :---: | :---: |:---: | :---: |
> | test ppl ($\downarrow$) | 408.62 | 187.25 | **169.98** | 4874.87
>
> The (0.3,0.9) parameter configuration exhibits catastrophic performance degradation (PPL >4800), producing semantically incoherent outputs that align with its poor performance across multiple tasks in our main experiments. This consistency underscores our method's robustness in identifying non-viable schedules across different generation paradigms.
>
> We hope that our experiment has addressed your problem and further discussion is welcomed.
>
> [1*] Qwen Team. "Qwen2.5: A Party of Foundation Models". 2024.
>
> ### Question 3:
>
> > In both subfigures at Figure 2 there are quite sharp bound after which energy start to drastically grow (green to purple edge). Why is that?
>
> Our computational analysis reveals that schedules deviating significantly from theoretical optima may induce the energy functionals to dirverge. The Beta distribution parameters systematically govern critical schedule characteristics including concavity-convexity properties and inflection point locations. When these parameters stray beyond critical thresholds from their optimal configurations, the energy functionals will may diverge to infinity.
>
> Our large-scale experiments corroborate this phenomenon: certain schedule may exhibits catastrophic performance degradation on specific tasks while maintaining normal functionality on others. This bifurcation originates from the energy landscape's acute sensitivity to $\gamma_t$ variations. Different $\gamma_t$ schedules create distinct energy topographies where non-optimal configurations reside in high-energy areas, leading to suboptimal generation performance.
>
> ### Question 4:
>
> > In Figure 2 caption the following said “Axes represent the beta-parameterization of $\alpha_t$ (see Sec. 3.3)”. What exactly does it mean, what is beta-parameterization of $\alpha_t$ ? I could not find such information in Sec. 3.3
>
> Thank you for highlighting this presentation gap. The beta-parameterization refers to constructing $α_t$ schedules through the Beta CDF: $α_t = CDF(\mathcal{B}(a,b))(t)$, where $(a,b)$ parameterize the Beta distribution. Figure 2 illustrates how these parameters influence energy landscapes under fixed $\gamma_t$ schedules through Condition 3.4. Each axis coordinate corresponds to specific $(a,b)$ pairs that generate distinct $α_t$ via this CDF mapping. The visualization demonstrates how Condition 3.4-guided parameter selection identifies energy minima. We will clarify this methodology explicitly in revisions to prevent misinterpretation.
>
> ### Question 5:
>
> > In Def 2.1 the unmasking process rate matrix is used, while as I understand it there also could be masking process rate matrix, am I right? If that is the case and in general these energy formulations (Def 2.1, Def 2.2, Def 2.3) can be defined in both ways (forward and backward) I think it should be mentioned somewhere in the main part of the paper.
>
> Thank you for your suggestion. While energy formulations could indeed be constructed for both masking and unmasking processes theoretically, our framework deliberately focuses on the unmasking rate matrix since the unmasking process directly governs the practical sampling trajectory – the primary optimization target for real-world applications.
>
> We hope that our response above has addressed your concerns, and we welcome further discussion if needed.

---

> > ### Comment · Reviewer_gvvr · 2025-08-05
> >
> > I thank the authors for taking the time to answer my questions. I’m mostly satisfied with the answers. Please incorporate additional clarifications and changes from the rebuttal answer into the final version of the paper. Then I would like to highlight several points from your answer.
> >
> > Regarding Weaknesses 2 & 3, I acknowledge your point that parametrization of $\gamma_t$ is derived from the energy perspective and hence presents a meaningful schedule. However, still from the partitioner's point of view, I feel like experimental proof of your point that constructing a schedule in the space of $\gamma_t$ is more effective than constructing the schedule in the space of $\alpha_t$ is missing from the paper. If indeed your point is true, then by optimization over $\alpha_t$ schedule, you should get far worse results than in the case of optimizing  $\gamma_t$.
> >
> > Regarding Weakness 5 and Question 2. I am satisfied with the answer and glad to see that you would incorporate these details into Section 4.
> >
> > That said, I would like to raise the score up to 4. But still, for me practical and theoretical parts of the paper do not go hand in hand. Unfortunately, it is not clear wherever $\gamma_t$ is needed in practice, because there is no comparison with the bold $\alpha_t$ parametrization or other schedule constructing methods not involving $\gamma_t$ parameterization. In my view, the primary strength of the paper lies in its practical contributions, whereas the theoretical component appears somewhat unconvincing.

---

> > > ### Author Response · Authors · 2025-08-06
> > >
> > > We appreciate the constructive suggestion. We will add an extended comparative analysis between $\alpha_t$ and $\gamma_t$ in the final version. We will also incorporate explicit acknowledgment that it is the introduction of $\gamma_t$ that motivated our Beta-CDF search methodology.
> > >
> > > We are glad that most of your concerns have been addressed. Thank you for acknowledging our contributions and updating your evaluation.

---

> ### Comment · Reviewer_2RNf · 2025-08-05
>
> I thank the authors for answering my questions.  They have addressed most of my concerns. I would like to raise my rating from 4 to 5.

---

### Official Review · Reviewer_2RNf · 2025-06-29

**Clarity:** 2
**Significance:** 2
**Originality:** 3
**Rating:** 5
**Confidence:** 3

**Summary:**

In this paper, the authors worked on Masked diffusion models(MDM) for discrete data. They proposed  to define three different energy formulations—kinetic, conditional kinetic, and geodesic energy, and then prove they are mathematically equivalent, thus minimize all three with the same  optimal sampling schedules that based on a Beta-CDF parameterization which they believe enables efficiency.

**Questions:**

The paper is difficult to read. The following are some of my questions:
1. It seems there is some mistake when the authors introduced the MDM. Page 2: line 62: The authors mentioned: "a masking process (operating backward in time, t = 1 → 0) and its reverse, the unmasking process (t = 0 → 1) ...", this seems opposite to what was in the previous literature where forward diffusion is masking and backward is unmasking. As a result,  in  Line 57: The authors assume p0(z) follows a uniform distribution, but they did not specify what is the target distribution p_1(z). Again, this contradicts with existing literatures. For example, ref[8] of the paper discussed the forward diffusion to a uniform distribution; and ref [43] works on the case that x_1 follows a Dirac distribution.

2. Page2: Eqns. (1) (2) did not make sense with (z not equal to x) in between the two lines, seems like a mistake.

3. The paper did not explicitly mentioned what the target distribution is; Figure 4 showed some, but did not explicitly explained it.

**Ethical Concerns:**

["NO or VERY MINOR ethics concerns only"]

**Final Justification:**

Thanks to the authors answering my questions. I would like to raise my rating to 5.

**Limitations:**

Yes.

**Paper Formatting Concerns:**

No.

**Quality:**

2

**Strengths And Weaknesses:**

It is interesting to see they proposed energy functions for characterizing MDM; however, the optimal sampling schedule defined to minimize the  the three formulations that the authors proposed sounds not to improve anything because the authors also showed the "Invariance of Training Loss to the Mask Schedule" in Appendix C. So, I wonder if the three are equivalent formation of the same concept.

Also, in the three definitions, what if the denominator, say in the eqn. (5), equals 0? I am further concerned when I read Page6, Figure 2: according to the pictures, the color indicates energy values, it seems like discontinuous for Beta(a,b). This suggested the energy function defined did not consider possible 0 in the denominator.

While authors focused on Q_t(z, x), I suggest they should provide p_t(z, x) when defining MDM. Also, they use the model in ref[43] for experiments; Further, it seems there is some issue with their definition of MDM sees not the same, see the section of Questions.

---

> ### Author Rebuttal · Authors · 2025-07-31
>
> Thank you for the recognition of our contributions and the thoughtful comments. Below is our point-by-point response.
>
> ### Weakness 1:
>
> > However, the optimal sampling schedule defined to minimize the the three formulations that the authors proposed sounds not to improve anything because the authors also showed the "Invariance of Training Loss to the Mask Schedule" in Appendix C. So, I wonder if the three are equivalent formation of the same concept.
>
> We appreciate this important question. The three energy formulations proposed in our work quantify the **sampling** process's "energy cost" from distinct perspectives, which fundamentally differ from the **training** loss function. To be more specifically, The training phase aims to learn a neural network approximation of the target distribution whereas the sampling phase addresses the operational trajectory for sample generation from the learned distribution.
>
> Therefore, while Appendix C demonstrates training loss invariance to mask schedules – enabling post-training schedule tuning – the sampling process remains highly schedule-dependent, as evidenced by our empirical results.
>
> We will adapt the above clarification in the main text of the final version to make our theoretical framework clearer.
>
> ### Weakness 2:
>
> > Also, in the three definitions, what if the denominator, say in the eqn. (5), equals 0?
>
> We thank you for this meticulous observation. **We would like to clarify that such cases are natually excluded from the summation, therefore does not affect our framework.** Consider the kinetic energy definition:
>
> $$\mathcal{E}_ k(p_t,Q_t;\gamma_t)=\mathbb{E}_ {t,p_t(z)}\sum_ {x:x\neq z}\frac{1}{\dot{\gamma}_tp_t(x)}Q_t(z,x)^2.$$
>
> To calculate this formulation, we first sample $z$ that have positive probability, i.e. $p_t(z)>0.$ Due to the detailed balance condition, as long as $Q_t(z,x)>0$, we must have $p_t(x)>0$ due to the continuity of the probability distribution. Therefore, the case $p_t(x)=0$ can only occur when $Q_t(z,x)=0$.
>
> Since the summation is of all probability flows between states, such $(z,x)$ pairs carrying zero transition probability ($Q_t(z,x) = 0$) are inherently and natually excluded from the summation, as it suggests that the flow between $z$ and $x$ does not exist at all. Our theoretical framework and proofs implicitly incorporate this exclusion principle throughout the analysis. We acknowledge the need to explicitly state this detail and we will revise accordingly in our final version in caption to enhance clarity. However, such concern does not affect our theoretical framework. Thank you for highlighting this important presentation consideration.
>
> ### Weakness 3:
>
> > I am further concerned when I read Page6, Figure 2: according to the pictures, the color indicates energy values, it seems like discontinuous for Beta(a,b). This suggested the energy function defined did not consider possible 0 in the denominator.
>
> We appreciate your careful examination. The apparent discontinuity in Figure 2 stems from our discrete parameter sampling strategy for visualization purposes rather than inherent energy function discontinuity. Energy values actually vary continuously with respect to $(a,b)$ parameters. Figure 2 is to emphasize the relationship between different weight functions and corresponding energy minima positions, therefore we only selected sparsely distributed $(a,b)$ pairs to show the coarse relationship. As addressed in our previous response regarding denominator zeros, this potential concern does not affect the theoretical validity or experimental results presented.
>
> ### Question 1:
>
> > It seems there is some mistake when the authors introduced the MDM. Page 2: line 62: The authors mentioned: "a masking process (operating backward in time, t = 1 → 0) and its reverse, the unmasking process (t = 0 → 1) ...", this seems opposite to what was in the previous literature where forward diffusion is masking and backward is unmasking. As a result, in Line 57: The authors assume p0(z) follows a uniform distribution, but they did not specify what is the target distribution p_1(z). Again, this contradicts with existing literatures. For example, ref[8] of the paper discussed the forward diffusion to a uniform distribution; and ref [43] works on the case that x_1 follows a Dirac distribution.
>
> Thank you for this insightful observation. Our temporal parameterization follows conventions from energy-based sampling literature [1*,2*] rather than standard diffusion frameworks. Specifically: $t=0$ denotes the initial sampling state (fully masked Dirac distribution in MDM) and $t=1$ marks the final generated state.
>
> This orientation aligns the forward temporal direction with the unmasking process – the core focus of our energy analysis. The reversed timeline offers two key advantages: 1) It simplifies energy functional expressions by synchronizing time progression with generation advancement, and 2) Ensures mask/interpolation schedules naturally increase from 0 to 1. While differing from some diffusion literature, this convention maintains internal consistency and better serves our analytical framework's objectives.
>
> [1*] Neta Shaul et al. “On Kinetic Optimal Probability Paths for Generative Models”. In: International Conference on Machine Learning, ICML 2023, 23-29 July 2023, Honolulu, Hawaii, USA. Ed. by Andreas Krause et al. Vol. 202. Proceedings of Machine Learning Research. PMLR, 2023, pp. 30883–30907.
>
> [2*] Neta Shaul et al. “FlowMatching with General Discrete Paths: A Kinetic-Optimal Perspective”. In: CoRR abs/2412.03487 (2024). DOI: 10.48550/ARXIV.2412.03487. arXiv: 2412.03487.
>
>
> ### Weakness 4 & Question 3:
>
> > While authors focused on Q_t(z, x), I suggest they should provide p_t(z, x) when defining MDM.
> >
> > The paper did not explicitly mentioned what the target distribution is; Figure 4 showed some, but did not explicitly explained it.
>
> We appreciate this technical clarification. In continuous-time Markov chains, the marginal distribution p_t(z) evolves via the Kolmogorov forward equation $\frac{dp}{dt} = p_tQ_t$. For MDM's specific $Q_t(z,x)$ structure, $p_t(z)$ generally lacks closed-form solutions but remains fully determined through $Q_t$ specifications. Therefore following MDM traditions [1*,2*], we only specify $Q_t(z,x)$ and $p_t(z)$ is then fully determined.
>
> As for the target distribution, our main experiments focus on practical scenarios where target distributions correspond to task-specific data distributions without explicit analytical forms. As for the synthetic experiment in Figure 4, the target distributions indeed have analytical forms and details are presented in Appendix E.4.
>
> [1*] Jacob Austin et al. “Structured denoising diffusion models in discrete state-spaces”. In: Advances in Neural Information Processing Systems 34 (2021), pp. 17981–17993.
>
> [2*] Jingyang Ou et al. “Your Absorbing Discrete Diffusion Secretly Models the Conditional Distributions of Clean Data”. In: ICLR. 2025.
>
> ### Question 2:
>
> > Page2: Eqns. (1) (2) did not make sense with (z not equal to x) in between the two lines, seems like a mistake.
>
> Thank you for this meticulous observation. The $z \neq x$ condition rigorously reflects continuous-time Markov chain conventions [1*]. Off-diagonal elements $Q_t(z,x) (z \neq x)$ & represent transition rates between distinct states, while diagonal elements satisfy $Q_t(z,z) = -\sum_{x\neq z}Q_t(z,x)$ to conserve probability mass. Explicitly specifying $z \neq x$ ensures mathematical precision when defining transition dynamics, as diagonal elements derive from off-diagonal terms rather than being independently specified.
>
> [1*] Jingyang Ou et al. “Your Absorbing Discrete Diffusion Secretly Models the Conditional Distributions of Clean Data”. In: ICLR. 2025.
>
> We hope that our response above has addressed your concerns, and we welcome further discussion if needed.

---

### Official Review · Reviewer_5LPg · 2025-07-02

**Clarity:** 3
**Significance:** 2
**Originality:** 2
**Rating:** 4
**Confidence:** 3

**Summary:**

The paper proposes a unified theoretical framework that interprets masked diffusion models (MDMs) as solutions to discrete optimal transport problems. Within this framework, three energy formulations—kinetic, conditional kinetic, and geodesic—are shown to be mathematically equivalent in the sense that their corresponding losses are proportional under the MDM structure. Leveraging this insight, particularly the geodesic perspective, the paper introduces an optimal mask schedule with analytically derived coefficients along the diffusion trajectory. This leads to practical sampling improvements while deepening the theoretical understanding of MDMs. Experiments on both synthetic and real-world benchmarks demonstrate that the proposed energy-inspired schedules consistently outperform hand-crafted baselines, particularly in low-step sampling regimes.

**Questions:**

Please refer to the Weaknesses section above for Questions.

**Ethical Concerns:**

["NO or VERY MINOR ethics concerns only"]

**Final Justification:**

I thank the authors for their responses. My concerns have been addressed, and I am happy to revise my evaluation from 3 to 4.

**Limitations:**

Please refer to the Weaknesses section above for Limitations.

**Quality:**

3

**Strengths And Weaknesses:**

Overall, the main strength of this paper lies in its unified theoretical framework for understanding MDMs through kinetic, conditional kinetic, and geodesic formulations. Additionally, the proposed masking schedule is well-motivated, as it aligns with the geodesic properties established by the theoretical analysis. Finally, the schedule also demonstrates practical improvements in MDM sampling performance. However, I list some potential weaknesses of the paper below:
- Although the study introduces a kinetic viewpoint, the proposed masking schedule relies heavily on geodesic insights already explored in [1]. This makes the kinetic formulation feel more like a minor addition, with the main contribution closely following prior work—thereby limiting the novelty of this work.
- The use of the Beta CDF as the masking schedule $\gamma$ is reasonable, as it generalises existing schedules such as linear and squared sine. It also enables inference-time optimisation in MDMs. However, the choice of the Beta distribution parameters $a, b$ appears to be heuristic. Could the authors elaborate on how to systematically search for or optimise $a, b$ rather than relying on trial and error? Additionally, how can varying $a, b$ values lead to improvements over the linear and squared sine schedules in certain scenarios (especially in low-step sampling regimes)? Any insights or empirical observations would be appreciated.

In my view, the unified theoretical framework appears somewhat disconnected from the proposed schedule. The most practical contribution lies in the new masking schedule, which is primarily inspired by prior work—limiting the overall novelty. Furthermore, the choice of parameters $a, b$ in the Beta distribution across different settings lacks sufficient justification. I currently lean towards a borderline reject, though I may revise my evaluation depending on the authors’ responses.

[1] Jaehyeong Jo and Sung Ju Hwang. Continuous Diffusion Model for Language Modeling. 2025.

**[After Rebuttal] I thank the authors for their responses. My concerns have been addressed, and I am happy to revise my evaluation from 3 to 4.**

---

> ### Author Rebuttal · Authors · 2025-07-31
>
> Thank you for your careful review and thoughtful comments. Below is our point-by-point response.
>
> ### Weakness 1:
>
> > Although the study introduces a kinetic viewpoint, the proposed masking schedule relies heavily on geodesic insights already explored in [1]. This makes the kinetic formulation feel more like a minor addition, with the main contribution closely following prior work—thereby limiting the novelty of this work.
> >
> > The unified theoretical framework appears somewhat disconnected from the proposed schedule. The most practical contribution lies in the new masking schedule, which is primarily inspired by prior work—limiting the overall novelty.
>
> We appreciate this critical perspective. We would like to clarify that **Our framework fundamentally operates within the optimal transport paradigm for efficient sampling beyond the geodesic perspective.** As optimal transport connects directly with efficient sampling [1*,2*], we establish provably optimal schedules for sampling efficiency in masked diffusion models. More importantly, unlike DFM's decoupled optimization of $p_t$ and $Q_t$, MDM's unique architecture – where both $p_t$ and $Q_t$ are jointly determined by $\alpha_t$ – natually raises fundamental questions about  whether such a constrained setting still can reach optimality. Our analysis demonstrates that MDM intrinsically encodes optimal rate matrices within its constrained structure. This theoretical guarantee strengthens confidence in MDM's elegant framework, particularly given its empirical success. **Such a perspective is absent in [1].**
>
> **Technically, our core objective of our analysis is to minimize the kinetic and conditional kinetic energy, which cannot be interpreted easily within the geodesic perspective.** Therefore, the geodesic method only gives us a mathemetical tool rather than a core goal. By utilizing the geodesic perspective, our main theorems are successfuly proven and understanding of MDM's capabilities are deepened. As noted by Reviewer dGWj, such synthesis "presents a fresh combination of concepts, extending prior research in non-trivial ways."
>
> **Even within the geodesic perspective, it is unclear how the pure geometric method in [1] can be used to prove the optimality of Condition 3.4.** By introducing the geodesic energy, our proof of Lemma 3.5 is proved using functional minimization methods, not only extending the original proof, but also successfully connected to our main goal in optimal transport.
>
> Therefore, although the closed-form condition was hinted in [1], our theoretical contributions beyond prior work is still non-trivial. We will clarify these points in the revised version and we hope that this response has addressed your question.
>
> [1*] Cédric Villani. Topics in optimal transportation. Vol. 58. American Mathematical Soc., 2021.
>
> [2*] Neta Shaul et al. “On Kinetic Optimal Probability Paths for Generative Models”. In: International Conference on Machine Learning, ICML 2023, 23-29 July 2023, Honolulu, Hawaii, USA. Ed. by Andreas Krause et al. Vol. 202. Proceedings of Machine Learning Research. PMLR, 2023, pp. 30883–30907.
>
> ### Weakness 2:
>
> > The use of the Beta CDF as the masking schedule $\gamma$ is reasonable, as it generalises existing schedules such as linear and squared sine. It also enables inference-time optimisation in MDMs. However, the choice of the Beta distribution parameters $a, b$ appears to be heuristic. Could the authors elaborate on how to systematically search for or optimise $a, b$ rather than relying on trial and error?
> >
> > The choice of parameters $a, b$ in the Beta distribution across different settings lacks sufficient justification.
>
> We appreciate this insightful question. Our theoretical analysis suggests that different downstream tasks may require generated text to possess specific intrinsic structures, which in turn necessitates the generation process to emphasize particular temporal phases. These temporal preferences are captured through different $\gamma_t$ schedules, which induce corresponding optimal $\alpha_t$ through Condition 3.4.
>
> Therefore, we hypothesize that the schedule preferences are mostly inherent to task nature rather than data specifics. To validate this hypothesis, we conducted the following experiments demonstrating that **randomly chosen small subsets (50-150 instances) of test data suffice for reliable schedule selection.** Specifically, we compare schedule performance between small test subsets and full evaluations (generation length=256, steps=64):
> | Beta-CDF Parameters | (0.5,0.5) | (1,1) | (0.9,0.3) | (0.3,0.9) |
> | :--- | :---: | :---: |:---: | :---: |
> | **task=GSM8K($\uparrow$)** |||||
> | Random subset 1 (n=132)| **56.06**  | 53.79 | 52.27 | 0.00 |
> | Random subset 2 (n=132)| **55.30**  | 52.27 | 50.76 | 0.76 |
> | Full test set (n=1319) | **52.69** | 50.72 | 51.86 | 0.45 |
> | **task=humaneval($\uparrow$)** |||||
> | Random subset 1 (n=82) | 17.07 | 35.37 | **43.90** | 3.66 |
> | Random subset 2 (n=82)| 6.10 | 15.85 | **19.51** | 2.44 |
> | Full test set (n=164) | 11.59 | 22.56 | **24.39** | 1.83 |
>
> While HumanEval evaluations exhibit greater variance due to smaller test populations (n=164 total), the relative performance rankings remain consistent across subsets - a critical indicator of our method's robustness. This empirical validation confirms that schedule preferences are very probably induced by intrinsic task attributes rather than specific data instances.
>
> Therefore in practice, we recommend conducting initial grid searches using small random test data subsets (about 50~150 instances) across parameters $a,b \in \\{0.1, 0.2, ..., 1.0\\}$ to find a set of well-performed schedules. After the coarse search, we can perform finer-grained selection on shortlisted candidates using larger data subsets. This approach achieves comprehensive parameter space exploration while maintaining computational feasibility.
>
> We will expand this discussion in Section 4 to better guide practitioners in adapting our methodology. We appreciate this valuable question that helps clarify implementation details.
>
>
> ### Weakness 3:
>
> > Additionally, how can varying $a, b$ values lead to improvements over the linear and squared sine schedules in certain scenarios (especially in low-step sampling regimes)? Any insights or empirical observations would be appreciated.
>
> Thank you for this insightful question. As mentioned in the response to Weakness 2, different intrinsic structures of tasks may induce different weight function $\gamma_t$, therefore inducing different $\alpha_t$ preferences. Since $\dot{\gamma}_t$ appears in the denominator of our energy functional expression, temporal phases where $\gamma_t$ has larger derivatives get less weight and phases where $\gamma_t$ is relatively flat get more focused. By varying the parameter $(a,b)$, we fundamentally alter the schedule's concavity and inflection points, therefore changing the temporal focus. When the temporal focus is aligned with intrinsic demand of the specific task, sampling performance is thereby promoted.
>
> Take the first toy experiment in Figure 4 as an illustrative case. Since the target distribution only have positive probability on extreme points (all zeros or all ones), precise trajectory control becomes paramount in initial generation phases to prevent simultaneous 0-1 generation. As the generation proceed to the ending period, however (for example 4 zeros have already been unmasked), the distribution itself (which is parameterized by a neural network in real world experiments) will prohibit the generation of ones since it will lead to invalid combinations. This eliminates the need for intensive late-stage focus, enabling efficient resource allocation to the decisive initial phases. Such temporal preference make the model prefer the parameter (0.6,0.3), where the starting period of generation is more weighted.
>
> In low-step regimes, strategic allocation of limited computational resources becomes more critical. This explains why our tuning method lead to improvements especially under low step sampling scenarios.
>
> We hope that our response above has addressed your concerns, and we welcome further discussion if needed.

---

### Official Review · Reviewer_nSvp · 2025-07-02

**Clarity:** 3
**Significance:** 3
**Originality:** 4
**Rating:** 5
**Confidence:** 2

**Summary:**

This paper presents a theoretical framework interpreting Masked Diffusion Models (MDMs) as minimizing energy functionals rooted in discrete optimal transport. It establishes that three distinct formulations of transport cost—kinetic energy, conditional kinetic energy, and geodesic energy—are mathematically equivalent within the MDM structure. A key result is the derivation of a closed-form optimal mask schedule, showing that MDMs minimize these energies under specific conditions.

**Questions:**

I only have one questions: the proposed method demonstrates strong improvements in few-step sampling on HumanEval and Hendrycks Math. However, the gains are less significant or inconsistent on other benchmarks like BBH and GSM8K. Can you elaborate on why the proposed energy-based schedule performs better on some tasks than others?

**Ethical Concerns:**

["NO or VERY MINOR ethics concerns only"]

**Final Justification:**

All of my concerns have been addressed and so I raised my score.

**Limitations:**

Yes.

**Quality:**

4

**Strengths And Weaknesses:**

Strengths: The paper offers a highly novel theoretical interpretation of MDMs, unifying different energy functionals and drawing on principles from information geometry and optimal transport. A major contribution is the identification of a closed-form condition that governs optimal masking schedules, bridging discrete probability flows and geodesic trajectories. The method is tested on strong baselines, including LLaDA 8B, showing significant improvements in low-step sampling efficiency on tasks like HumanEval and Hendrycks Math.

weaknesses: While experiments show improved sample efficiency, the performance gains are limited to a subset of benchmarks (notably, not significant on BBH or GSM8K).

---

> ### Author Rebuttal · Authors · 2025-07-31
>
> Thank you for recognizing our contributions and providing thoughtful feedback. Please find our response below.
>
> ### Weakness & Question:
>
> > While experiments show improved sample efficiency, the performance gains are limited to a subset of benchmarks (notably, not significant on BBH or GSM8K)
> >
> > The proposed method demonstrates strong improvements in few-step sampling on HumanEval and Hendrycks Math. However, the gains are less significant or inconsistent on other benchmarks like BBH and GSM8K. Can you elaborate on why the proposed energy-based schedule performs better on some tasks than others?
>
> We appreciate this insightful question. We clarify that our methodology provides both a theoretical foundation for understanding schedule preferences and a practical mechanism for task-specific optimization, rather than proposing a universally superior schedule.
>
> **In fact, linear schedule $\alpha_t = t$ is a special case in our framework since it corresponds to $\gamma_t=CDF(\mathcal{B}(0.5,0.5))$, a point in our parameter space.** Therefore, the empirical parity on certain benchmarks confirms that linear schedules already serve as near-optimal candidates for specific task categories. However, our framework's value lies in enabling systematic task-specific schedule optimization, overcoming the limitations of previous one-size-fits-all approaches.
>
> Specifically, the linear schedule $\alpha_t = t$ corresponds to $\gamma_t=CDF(\mathcal{B}(0.5,0.5))$, which has higher derivatives on both initial and final generation phases while being relatively static in the middle. Since $\dot{\gamma}_t$ appears in the denominator of our energy functional expression, this implies that tasks requiring sustained refinement throughout generation (particularly middle phases) rather than the starting or ending phases might inherently favor the linear schedule.
>
> Among the six tasks we experimented on, BBH is only one that focuses on general reasoning problems. As for GSM8K, although it is mathematics-focused, its prompt requires the generated text to use detailed natural language explanations compared to other math benchmarks. This might be the reason why sustained refinement in the middle phases are preferred. Considering this similarity, it is reasonable that GSM8K and BBH have similar preference on schedules although it is difficult to mathematically deduce the exact expression of the energy functionals.
>
> Considering the complexity of real-world tasks, it is indeed a challenging direction to systematically study the relationship between tasks and their prefered schedules. A promising research direction involves designing intermediate benchmarks that balance practical relevance with analytical tractability - more sophisticated than our current toy examples yet simpler than real-world tasks.
>
> We will incorporate this analysis in Section 4 of our paper in the final version. We appreciate this insightful question and we hope our response addresses your concerns. Further discussion is welcomed if needed.

---

> > ### Comment · Reviewer_nSvp · 2025-08-06
> >
> > Thank you for the response. My concerns have been addressed and so I have raised my score.

---

> > > ### Author Response · Authors · 2025-08-06
> > >
> > > Thank you for updating your evaluation. We are glad that your concerns have been addressed.

---

### Official Review · Reviewer_dGWj · 2025-07-02

**Clarity:** 3
**Significance:** 3
**Originality:** 4
**Rating:** 6
**Confidence:** 3

**Summary:**

The submission proposes a theoretical framework for Masked Diffusion Models (MDM) based on energy minimization in optimal transport and from that derives an optimal sampling schedule. Kinetic energy, conditional kinetic energy, and geodesic energy are shown to be equivalent under the MDM formulation, which connects Kinect energy as a sampling-efficiency view with a geometric interpretation (geodesic paths) in discrete diffusion. The authors derive a closed-form condition for the optimality of the mask schedule $alpha_t$, namely $alpha_t^* = \sin^2(\pi/2\cdot \gamma_t)$, where $\gamma_t$ is a continuous interpolation schedule. For this condition, the MDM minimizes all three energy functionals simultaneously, i.e. MDMs achieve optimal transport trajectories when the condition holds.
From that theoretical result, the authors derive a technique to tune mask schedules after training, by parameterizing $\gamma_t$ as the CDF of a Beta distribution with two parameters, which allow for search of a near-optimal schedule in 2D without retraining of the model. This is possible, because MDM training objectives are shown to be invariant to the chosen schedule. Experiments on synthetic data and real benchmarks confirm that the optimized schedules yield faster sampling without a reduction in output quality. For example, on a code generation task, the optimal schedule matches the quality of a linear schedule using only half the number of denoising steps. Likewise, on a math reasoning task, it reached qualitative parity with a quarter of the steps, showing improved few-step generation performance.

**Questions:**

- The theorems assume a continuous-time interpolation from 0 to 1. In practice, a discrete sequence of masks must be chosen. Is the sine-squared schedule still exactly optimal in a discretized sense, or only in the continuous limit? Currently this is only done implicitly by tuning Beta parameters for a fixed step count. A brief clarification of the theoretical connection would be helpful.

- How are the parameters of the Beta-CDF selected for a given task? Did you perform a grid search on (a, b) and pick the best performing schedule? If yes, what are the parameters of this search? It would be useful to know the number of evaluations needed and whether it is feasible to apply to very large models. Discussing the tuning strategy would also help readers to adapt your method more easily and extend this part of it.

- Do you have hypotheses why GSM8K and BBH did not benefit from optimized Beta schedules? Is it because those tasks require very specific refinement at the end, so a linear schedule is already as good as it gets? Providing some discussion on these cases would be valuable.

**Ethical Concerns:**

["NO or VERY MINOR ethics concerns only"]

**Final Justification:**

I appreciate the detailed responses from the authors, which answered all my questions and concerns. I am happy to raise my rating to “strong accept.”

**Limitations:**

yes

**Paper Formatting Concerns:**

no formatting issues

**Quality:**

4

**Strengths And Weaknesses:**

Quality:
The work bases its applications on a thoroughly derived mathematical foundation. The proofs appear sound and the central result, Theorem 3.6, in particular is non-trivial (minimizing geodesic energy does not guarantee minimizing kinetic energy) but the authors address this carefully in their argumentation.
Empirically, diverse benchmarks were chosen (MBPP, HumanEval for code, GSM8K and Math for reasoning) and the authors show clear evidence that the schedules found via Beta tuning significantly reduce sampling steps without reducing quality. The methodology is solid and supported by informative plots (e.g. Fig 5). In addition, showing tasks where their method does not outperform the baseline (GSM8K, BBH) goes a long way in presenting a complete picture, including the limitations. Enough details are given to ensure reproducibility.
While mathematically sound, the authors don’t explicitly discuss the connection between the assumed continuous-time diffusion setting in the theoretical part and the finite-step discrete sampling in real settings. For example, Theorem 3.6 proves the optimality in the continuum and some discussion of how closely the discretized schedule approximates the continuous optimum would strengthen the rigor of the derivation.

Clarity:
The paper is written clearly, despite its theoretical depth. The introduction motivates the problem and lays out the key points of the paper logically. Fig. 1 is helpful in both summarizing and guiding the reader through Section 3. Notation is consistent and deferring the heavy mathematical proofs to the appendix helps the main text stay focused.
While the notation is mostly standard, expressions like in the conditional kinetic energy definition (Eq. 6) might require readers that are not familiar with OT to study the appendix thoroughly before being able to follow the results and keeping track of the notation. In addition, some key ideas, like the mask schedule invariance of the ELBO objective are mentioned somewhat briefly and more emphasis on such points in the main text could improve the reader’s understanding of why post-training schedule tuning is possible.

Significance:
The paper addresses a significant gap in the understanding of MDMs, which prior to this work showed promising results, but used an ad hoc sampling process based on manually chosen mask schedules without theoretical justification. The contributions of this work significantly advancing the understanding by grounding the sampling of MDM in OT theory. This can guide future design of diffusion-based generative modes as it is possible to reason about MDM behavior using energy minimization and geodesics. And practically, the faster sampling is highly important for most downstream tasks. In addition, not requiring retraining is a big advantage for deployment, because pretrained models can be tuned cheaply for new tasks.
Nevertheless, the empirical performance gains are not universal, as some benchmarks showed no improvements over linear schedules. This shows that further research is needed to gain a better understanding, which the authors acknowledge as a future direction. Nonetheless, the work provides a good foundation for future research to build upon.

Originality:
The contributions of this work are highly novel. While prior work was drawing connections between DMs and OT, to my knowledge this is the first work to interpret MDMs as energy-minimizing transport processes. The closed-form optimal schedule condition was hinted at in earlier analysis (Jo & Hwang, 2025: Continuous Diffusion Model for Language Modeling), but it did not connect it to sampling energy minimization. The proposed Beta-CDF parameterization is a clever choice to simplify the schedule search and the authors show how known schedules are specific points in that space. Related work is cited clearly and the differences are made clear. While this area of research is very active, the paper presents a fresh combination of concepts, extending prior research in non-trivial ways.

---

> ### Author Rebuttal · Authors · 2025-07-31
>
> Thank you for your extremely thorough review and acknowledgment of our contributions. Below is our point-by-point response.
>
> ### Weakness (Quality) & Question 1:
>
> > While mathematically sound, the authors don’t explicitly discuss the connection between the assumed continuous-time diffusion setting in the theoretical part and the finite-step discrete sampling in real settings.
> >
> > The theorems assume a continuous-time interpolation from 0 to 1. In practice, a discrete sequence of masks must be chosen. Is the sine-squared schedule still exactly optimal in a discretized sense, or only in the continuous limit? Currently this is only done implicitly by tuning Beta parameters for a fixed step count. A brief clarification of the theoretical connection would be helpful.
>
> We appreciate this insightful question. Indeed, **under Riemann discretization, MDM trajectories still minimizes a corresponding discrete energy functional that converges to the continuous formulation as the number of steps $N \to \infty$.**
>
> Due to the equivalence of the three energy functionals, we focus our analysis on the geodesic energy. Recall that $\mathcal{E}_ g$ can be equivalently expressed as:
>
> $\mathcal{E}_ g = \int_{0}^1 \frac{||\dot{y}_t||^2}{\dot{\gamma_t}} dt.$
>
> Therefore, we can discretize $\mathcal{E}_g$ as:
>
> $\mathcal{E}_ g^N = \sum_{n=0}^{N-1} \frac{||y_{n+1}-y_{n}||^2 /(\Delta t)^2}{|\gamma_{n+1}-\gamma_{n}| /\Delta t} \Delta t = \sum_{n=0}^{N-1} \frac{||y_{n+1}-y_{n}||^2 }{|\gamma_{n+1}-\gamma_{n}|},$
>
> where $\Delta t=1/N.$ For simplicity, consider the special case where $\gamma_t = t$. Then we have $|\gamma_{n+1}-\gamma_{n}| = \frac{1}{N}$. We now demonstrate that the $N$ equidistant points along the geodesic still minimize this functional. First, we note that the squared chord length on the unit sphere satisfies $||y_{n+1}-y_{n}||^2 = 2(1-\cos(\theta_n))$, where $\theta_n$ denotes the angle between $y_n$ and $y_{n+1}$. This transforms the minimization problem into finding $N-1$ intermediate points along the great circle arc between $y_0$ and $y_1$ that minimize $\sum_n 2(1-\cos(\theta_n))$.
>
> From the properties of spherical geodesics, we know that placing all intermediate points along the geodesic path enforces the constraint $\sum\theta_n = \frac{\pi}{2}$. Given that $1-\cos(x)$ is convex over $(0, \pi/2)$, Jensen's inequality establishes:
>
> $\sum_n 2(1-\cos(\theta_n)) \geq N\cdot 2(1-\cos(\frac{\pi}{2N})),$
>
> with equality achieved when points are uniformly distributed along the geodesic. Any deviation from the geodesic path would further increase the cumulative angular separation $\sum\theta_n$ beyond $\frac{\pi}{2}$, consequently raising the total energy. This proof extends to arbitrary $\gamma_t$ schedules through proportional point distribution based on $|\gamma_{n+1}-\gamma_{n}|$.
>
> We will add the above discussion in Section 3.2 of the paper in the final version. We are grateful for this constructive question that allows us to strengthen the theoretical grounding of our approach.
>
> ### Weakness (Clarity):
>
> > While the notation is mostly standard, expressions like in the conditional kinetic energy definition (Eq. 6) might require readers that are not familiar with OT to study the appendix thoroughly before being able to follow the results and keeping track of the notation.
> >
> > In addition, some key ideas, like the mask schedule invariance of the ELBO objective are mentioned somewhat briefly and more emphasis on such points in the main text could improve the reader’s understanding of why post-training schedule tuning is possible.
>
> Thank you for the constructive suggestions. We will improve the presentation by providing clearer explanations of the definitions in the main text to enhance readability. Also, the ELBO's invariance to mask scheduling indeed plays a crucial role in our framework, therefore we will emphasize this fundamental property more prominently in the main text to improve conceptual clarity.
>
> ### Question 2:
>
> > How are the parameters of the Beta-CDF selected for a given task? Did you perform a grid search on (a, b) and pick the best performing schedule? If yes, what are the parameters of this search? It would be useful to know the number of evaluations needed and whether it is feasible to apply to very large models. Discussing the tuning strategy would also help readers to adapt your method more easily and extend this part of it.
>
> We appreciate this insightful question. Our theoretical analysis suggests that different downstream tasks may require generated text to possess specific intrinsic structures, which in turn necessitates the generation process to emphasize particular temporal phases. These temporal preferences are captured through different $\gamma_t$ schedules, which induce corresponding optimal $\alpha_t$ through Condition 3.4.
>
> Therefore, we hypothesize that the schedule preferences are mostly inherent to task nature rather than data specifics. To validate this hypothesis, we conducted the following experiments demonstrating that **randomly chosen small subsets (50-150 instances) of test data suffice for reliable schedule selection.** Specifically, we compare schedule performance between small test subsets and full evaluations (generation length=256, steps=64):
> | Beta-CDF Parameters | (0.5,0.5) | (1,1) | (0.9,0.3) | (0.3,0.9) |
> | :--- | :---: | :---: |:---: | :---: |
> | **task=GSM8K($\uparrow$)** |||||
> | Random subset 1 (n=132)| **56.06**  | 53.79 | 52.27 | 0.00 |
> | Random subset 2 (n=132)| **55.30**  | 52.27 | 50.76 | 0.76 |
> | Full test set (n=1319) | **52.69** | 50.72 | 51.86 | 0.45 |
> | **task=humaneval($\uparrow$)** |||||
> | Random subset 1 (n=82) | 17.07 | 35.37 | **43.90** | 3.66 |
> | Random subset 2 (n=82)| 6.10 | 15.85 | **19.51** | 2.44 |
> | Full test set (n=164) | 11.59 | 22.56 | **24.39** | 1.83 |
>
> While HumanEval evaluations exhibit greater variance due to smaller test populations (n=164 total), the relative performance rankings remain consistent across subsets - a critical indicator of our method's robustness. This empirical validation confirms that schedule preferences are very probably induced by intrinsic task attributes rather than specific data instances.
>
> Therefore in practice, we recommend conducting initial grid searches using small random test data subsets (about 50~150 instances) across parameters $a,b \in \\{ 0.1, 0.2, ..., 1.0 \\} $ to find a set of well-performed schedules. After the coarse search, we can perform finer-grained selection on shortlisted candidates using larger data subsets. This approach achieves comprehensive parameter space exploration while maintaining computational feasibility.
>
> We will expand this discussion in Section 4 to better guide practitioners in adapting our methodology. We appreciate this valuable question that helps clarify implementation details.
>
> ### Weakness (Significance) & Question 3:
>
> > Nevertheless, the empirical performance gains are not universal, as some benchmarks showed no improvements over linear schedules. This shows that further research is needed to gain a better understanding, which the authors acknowledge as a future direction.
> >
> > Do you have hypotheses why GSM8K and BBH did not benefit from optimized Beta schedules? Is it because those tasks require very specific refinement at the end, so a linear schedule is already as good as it gets? Providing some discussion on these cases would be valuable.
>
> Our theoretical analysis suggests that different tasks may prefer different schedules, therefore the observed parity with linear baselines in certain tasks aligns with our theoretical framework.
>
> Specifically, the linear schedule $\alpha_t = t$ corresponds to $\gamma_t=CDF(\mathcal{B}(0.5,0.5))$, which has higher derivatives on both initial and final generation phases while being relatively static in the middle. Since $\dot{\gamma}_t$ appears in the denominator of our energy functional expression, this implies that tasks requiring sustained refinement throughout generation (particularly middle phases) rather than the starting or ending phases might inherently favor the linear schedule.
>
> Among the six tasks we experimented on, BBH is only one that focuses on general reasoning problems. As for GSM8K, although it is mathematics-focused, its prompt requires the generated text to use detailed natural language explanations compared to other math benchmarks. This might be the reason why sustained refinement in the middle phases are preferred. Considering this similarity, it is reasonable that GSM8K and BBH have similar preference on schedules although it is difficult to mathematically deduce the exact expression of the energy functionals.
>
> Considering the complexity of real-world tasks, it is indeed a challenging direction to systematically study the relationship between tasks and their prefered schedules. A promising research direction involves designing intermediate benchmarks that balance practical relevance with analytical tractability - more sophisticated than our current toy examples yet simpler than real-world tasks.
>
> We will incorporate this analysis in Section 4 of our paper in the final version. We appreciate this insightful line of questioning and we hope our response addresses your concerns. Further discussion is welcomed if needed.

---

### Note · Authors · 2025-08-14

We sincerely thank the Senior Area Chairs and the Area Chairs for their efforts in organizing the conference, and we thank all reviewers for their insightful suggestions and recognition of our work. We are particularly pleased to have addressed most concerns raised by the reviewers, leading to improved evaluations of our paper.

In summary, our work establishes a theoretical framework interpreting Masked Diffusion Models (MDMs) as optimal transport processes. By combining the geodesic and kinetic perspectives, we derive a closed-form condition for energy-optimal mask schedules, demonstrating that MDMs minimize sampling energy cost under this condition.

Furthermore, we introduce a Beta-CDF parameterization enabling efficient, task-adaptive schedule tuning within a 2D parameter space. Our theory is empirically validated across synthetic and real-world benchmarks, showing consistent improvements in few-step sampling performance.

During rebuttal, we not only provided more detailed explanations of our theoretical approach and significance, clarifying why different downstream tasks prefer distinct schedules within our framework, but also elaborated on the practical implementation of our Beta-CDF tuning method for identifying superior schedules. We also addressed some mathematical details and clarified certain descriptions. All modifications discussed will be incorporated into the final version.

Following the rebuttal, our work has received acknowledgment and approval from all five reviewers. We sincerely appreciate the reviewers' insightful questions and suggestions, which have significantly enhanced the quality of our work.

---

### Decision · Program_Chairs · 2025-09-17

**Decision:**

Accept (poster)

**Comment:**

This paper presents a theoretical framework that interprets Masked Diffusion Models (MDMs) through the lens of energy optimization in optimal transport. The authors demonstrate the mathematical equivalence of three distinct energy formulations - kinetic, conditional kinetic, and geodesic energy - within the MDM structure. A key theoretical contribution is the derivation of a closed-form optimality condition for the mask schedule, showing that MDMs minimize all three energies when this condition is satisfied. Building on this theory, the authors propose a practical method for schedule optimization by parameterizing the interpolation schedule via the CDF of a Beta distribution. Empirical results on synthetic and real-world benchmarks (e.g., code generation, math reasoning) demonstrate that their energy-inspired schedules can outperform standard hand-crafted baselines.

The topic of MDM models is currently of very high interest to the community. The paper received five reviews which, after discussion, all resulted in positive scores highlighting many aspects of the paper (novelty of the theory and interpretation, practical insights about schedule optimization, etc.). At the same time, one of the reviewers (gvvr) raised a valid and crucial point, questioning the profound theoretical significance of the work and suggesting that the theory might serve a primarily "decorative" role for what is essentially a practical contribution of a better schedule search method. After reading the paper, the meta-reviewer also agrees with this concern. Nevertheless, the meta-reviewer believes that the paper provides a valuable contribution to the community, with the merits (primarily the practical part) outweighing the flaws. The paper will be of interest to the community, and its acceptance is recommended.